# Morphologic determinant of tight junctions revealed by claudin-3 structures

Shun Nakamura[1,2], Katsumasa Irie[1,2], Hiroo Tanaka[3], Kouki Nishikawa[1], Hiroshi Suzuki [1,5], Yasunori Saitoh[1,6], Atsushi Tamura[3], Sachiko Tsukita[3] & Yoshinori Fujiyoshi[1,4]

Tight junction is a cell adhesion apparatus functioning as barrier and/or channel in the paracellular spaces of epithelia. Claudin is the major component of tight junction and polymerizes to form tight junction strands with various morphologies that may correlate with their functions. Here we present the crystal structure of mammalian claudin-3 at 3.6 Å resolution. The third transmembrane helix of claudin-3 is clearly bent compared with that of other subtypes. Structural analysis of additional two mutants with a single mutation representing other subtypes in the third helix indicates that this helix takes a bent or straight structure depending on the residue. The presence or absence of the helix bending changes the positions of residues related to claudin-claudin interactions and affects the morphology and adhesiveness of the tight junction strands. These results evoke a model for tight junction strand formation with different morphologies – straight or curvy strands – observed in native epithelia.

[1] Cellular and Structural Physiology Institute, Nagoya University, Furo-cho, Chikusa, Nagoya 464-8601, Japan. [2] Graduate School of Pharmaceutical Sciences, Nagoya University, Furo-cho, Chikusa, Nagoya 464-8601, Japan. [3] Laboratory of Biological Science, Graduate School of Frontier Biosciences and Graduate School of Medicine, Osaka University, Suita, Osaka 565-0871, Japan. [4] CeSPIA Inc., 2-1—1 Otemachi, Chiyoda, Tokyo 100-0004, Japan. [5] Present address: Laboratory of Molecular Electron Microscopy, The Rockefeller University, New York 10065, USA. [6] Present address: Research Institute for Interdisciplinary Science, Okayama University, Tsushima Naka 3-1—1, Kita, Okayama 700-8530, Japan. Correspondence and requests for materials should be addressed to Y.F. (email: yoshi@cespi.nagoya-u.ac.jp)

Cell adhesion is a fundamental mechanism for multicellular formation, and compartmentalization to various tissues and organs enables advanced life phenomena. Tight junction (TJ) is a cell adhesion apparatus found in vertebrates and urochordates, and is critical for the homeostasis by maintaining the internal environment of the highly complicated compartments. Each compartment is sealed by epithelial cell sheets, whose intercellular gaps are almost completely occluded by TJs[1]. TJs regulate substance permeation through the paracellular pathway as an impermeant or permselective barrier[2]. TJs are visualized by freeze-fracture electron microscopy as net-like architectures around apical membranes, called TJ strands[3]. Claudins (Cldns), tetra-span-transmembrane proteins, are the major essential components of TJ strands and are responsible for the barrier function[4,5]. Dysregulation or mutation of Cldn causes disfunction of TJ strands, resulting in numerous diseases[6,7].

TJ strands among different epithelia have different morphologies. The relationship between the morphology and properties of TJ strands is well discussed, but still elusive[8–10]. To evaluate the effect of TJ morphology on TJ properties, it is necessary to change the TJ morphology with minimal mutation in the same expression system. Anatomic observation of TJ strands shows that their morphology is related to the Cldn subtype[11,12]. The 27 Cldn subtypes identified to date in mammals[13] are expressed in an organ-specific and developmental stage-dependent manner[14–16]. TJ strands of each subtype exhibit different adhesion and paracellular permeation properties[17]. Cldns form TJ strands by two types of interactions: cis-interactions between Cldn molecules on the same cell membrane in a side-by-side manner for polymerization, and trans-interactions between Cldn molecules on opposing membranes through the extracellular parts in a head-to-head manner for cell–cell adhesion. Each subtype interacts with specific subtypes, not only homophilically, but also heterophilically[18,19]. As a result, combinations of the different subtypes with unique properties enable Cldns to form organ-specific TJ strands for the formation of optimized environments.

The crystal structure of mouse claudin-15 (mCldn15) exhibits cis-interactions in crystal packing[20]. Based on this structure, TJ strands are proposed to be formed by anti-parallel double-rows of Cldn polymers on the same membrane, and the paracellular pore is assumed to be achieved by formation of the porin-like β-barrel structure comprising the extracellular regions of the four Cldns coming from trans-interacting double-rows[21]. The structures of mouse claudin-19 (mCldn19) and human claudin-4 (hCldn4) were determined in complex with the C-terminal region of *Clostridium perfringens* enterotoxin (C-CPE)[22,23]. CPE, produced by bacteria that cause food poisoning, disrupts TJ strands[24], although C-CPE binding itself to Cldns without the N-terminal region is not cytotoxic[25]. The structures of Cldn/C-CPE complexes provide a model for the disassembly of Cldn–Cldn interactions by C-CPE binding. The structural basis of the subtype-specific properties arising diverse TJs, such as the various morphologies of TJ strands, is still elusive, however, due to the insufficiency of structural information from different Cldn subtypes.

## Results

**Structure of mCldn3.** To study this issue, we determined the structure of mouse claudin-3 (mCldn3) as an additional Cldn subtype among structure-unknown subtypes. Cldn3 was identified as one of the first membrane proteins of the CPE receptor, initially named RVP-1[26–28], and has therefore been studied for a long time. Cldn3 is expressed in a wide variety of epithelia, such as the gastrointestinal tract, urinary tract, and liver[14].

For the crystallization of mCldn3, 36 residues at the C-terminus were deleted and 4 membrane-proximal cysteines were substituted with alanines (Supplementary Figure 1a). We crystallized this construct (mCldn3$_{cryst}$) in complex with C-CPE carrying an S313A mutation, which enhances the thermostability of the complex[29]. The crystal structure was determined at a resolution of 3.6 Å by molecular replacement (Supplementary Table 1, Supplementary Figure 1b). The asymmetric unit contained two mCldn3/C-CPE complexes, which formed an upside-down dimer of the complex interacting through transmembrane helices (Supplementary Figure 2a, d, g, Supplementary Figure 3a). The electron density was well observed throughout the molecules, except for the amino acid residues at positions 66 to 70 in mCldn3, which correspond to the extracellular helix (ECH) in mCldn15.

The overall structure of mCldn3 is similar to that of the other subtypes for which the structures have been determined (Fig. 1a), and resembles the shape of a human left hand (Fig. 1b); the four transmembrane helices (TM1–TM4) form a helix bundle, which correspond to the forearm; the two extracellular segments, ECS1 (from β1 to β4) and ECS2 (from the extracellular part of TM3 to β5), include an anti-parallel β-sheet, and correspond to the four fingers and thumb. C-CPE binds to two loops of mCldn3—the

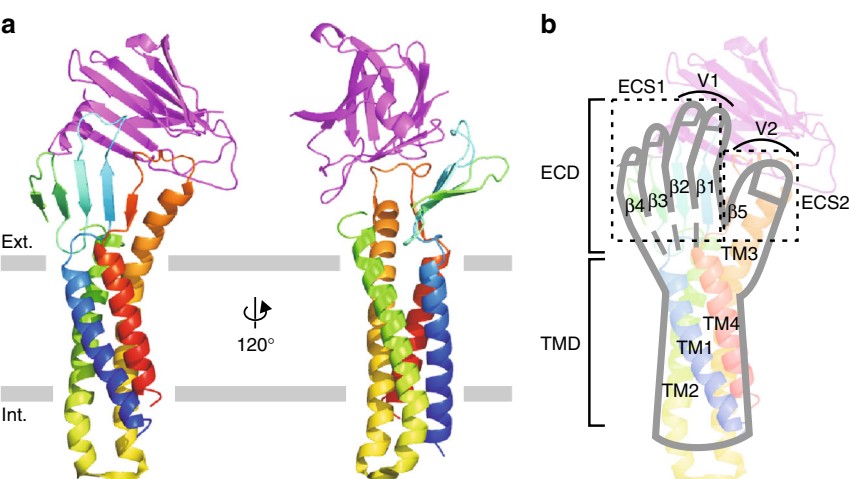

**Fig. 1** Structure of mCldn3 in complex with C-CPE. **a** Overall structure of the mCldn3$_{cryst}$/C-CPE complex in ribbon representation viewed parallel to the cell membrane. mCldn3$_{cryst}$ in rainbow colors from the N terminus (blue) to the C-terminus (red). C-CPE is colored magenta. Gray bars suggest boundaries of outer (Ext.) and inner (Int.) leaflets of the lipid bilayer. **b** Left-hand model of the mCldn3 structure

loop between β1 and β2 (V1 region) and the loop between TM3 and β5 (V2 region), in which residues related to C-CPE binding are at the same position as those of mCldn19 and hCldn4 (Supplementary Figure 4a, b). C-CPE, however, is located over the β-sheet of mCldn3 more distantly than those of mCldn19 and hCldn4 (Supplementary Figure 4c, e, f). Explaining with left-hand model, the hands of mCldn19 and hCldn4 grasp and enwrap C-CPE with the palms, whereas the hand of mCldn3 pinches C-CPE with the thumb and the index and middle fingers. Considering that mCldn3 binds with C-CPE with high affinity[22], the proximity of C-CPE to the β-sheet is not important for C-CPE binding. This different arrangement of the C-CPE is suggested to derive from the flexibility of the V1 and V2 regions, whose structures form loops and are not fixed. Mutational experiments revealed that substitution of Leu150 in mCldn3 for serine, the corresponding residue in Cldn19, markedly decreased C-CPE affinity (Supplementary Figure 6a). Conversely, leucine substitution of Ser152 in mCldn19 increased the affinity[22]. A side-chain of this residue protrudes into a hydrophobic pocket comprising tyrosines of the C-CPE (Supplementary Figure 6b) as previously predicted by a mutagenesis study[30], and the interaction in this position determines C-CPE affinity.

Structural superposition between mCldn3 and the other subtypes shows that the transmembrane domains (TMD) are identical, but the extracellular domain (ECD) of mCldn3 largely inclines as a whole toward the lipid bilayer in the direction of the back of the hand in the left-hand model, more than those of other subtypes (Fig. 2). In the ECD of the mCldn3 structure, the top part of TM3, which protrudes outside the membrane, bends near the boundary between the outer leaflet of the lipid bilayer and the extracellular side. Pro134 locates at the starting point of the TM3 bending, and the steric hindrance of its main chain possibly causes the TM3 helix to bend.

**Structures of two Pro134 mutants**. In the mammalian Cldn family, the residues corresponding to Pro134 are conserved, comprising mainly three amino acids; proline, glycine, and alanine, which occupy approximately 50%, 30%, and 20% of the position, respectively (Supplementary Figure 5b). While the structures of mCldn15 and mCldn19, which possess alanine at this position, do not show TM3 bending, the TM3 of hCldn4 with

proline is certainly bent (Fig. 2). The TM3 bending was predicted and recognized in previous studies[23,31,32], but its cause and effect were not clarified or comprehensively discussed. The mCldn3 structure suggests Pro134 is a key residue to determine subtype-specific overall structure. Therefore we focused on the residue at this position, and named it TM3 thenar because of its location at the thenar, which is the fleshy area of the palm at the base of the thumb, in the left-hand model (Fig. 3c).

To elucidate whether the TM3 bending is caused by the proline at the TM3 thenar and how it affects the overall structure, we further determined the crystal structures of two Pro134 mutants, mCldn3$_{cryst}$ P134G, and mCldn3$_{cryst}$ P134A, both in complex with C-CPE S313A at 4.3 Å and 3.9 Å resolution, respectively (Supplementary Table 1, Supplementary Figure 1c, d). The overall structures of the two mutants show that the TM3s of both mutants were straightened at the extracellular side as in the structures of mCldn15 and mCldn19, which have an alanine at the TM3 thenar (Fig. 3a). The other part of the TMD conformation and the Cldn-C-CPE interaction modes of the two mutants are identical to those of mCldn3$_{cryst}$ (Supplementary Figure 4), and the affinities to C-CPE were sufficiently comparable to that of wild-type mCldn3 (Supplementary Figure 6). The crystallization condition and crystal packing of mCldn3$_{cryst}$ P134A completely differed from that of mCldn3$_{cryst}$, whereas that of mCldn3$_{cryst}$ P134G was very similar to that of mCldn3$_{cryst}$ (Supplementary Figure 2, Supplementary Figure 3). Despite the same crystal packing between mCldn3$_{cryst}$ and mCldn3$_{cryst}$ P134G, the TM3 in the former was bent and the TM3 in the latter was straight. Therefore, the structures of the two mutants revealed that the TM3 bending of mCldn3 derives from the proline at the TM3 thenar, and not from C-CPE binding or crystal packing. Furthermore, the low resolution limit of the mCldn3$_{cryst}$ P134G crystal, despite the similar crystallization condition with mCldn3$_{cryst}$, suggested that the property of the glycine to destabilize helix conformation rendered the TM3 helix flexible. The B-factor analysis for each chain in an asymmetric unit also supports this notion, as described below. Comparison of the B-factors between the two Cldn molecules in the upside-down dimer (Supplementary Figure 2g) revealed that the B-factors of two molecules are similar to each other in the case of mCldn3$_{cryst}$ and mCldn3$_{cryst}$ P134A, whereas the B-factor of chain C is relatively higher than that of chain A only in the case of

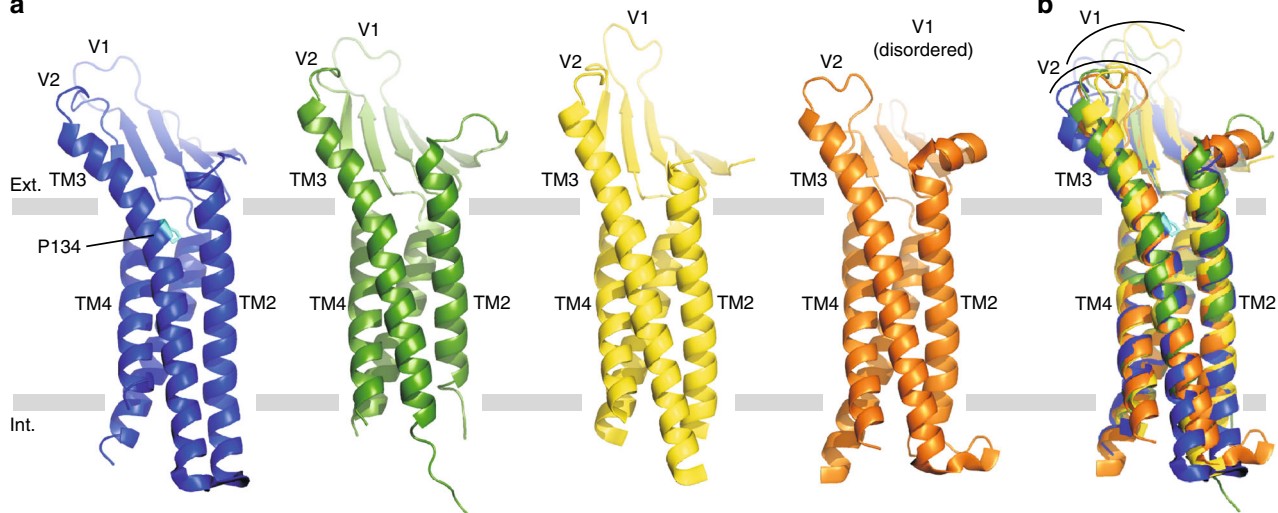

**Fig. 2** Comparison with structure-determined subtypes. **a** Structure-determined Cldn subtypes. mCldn3$_{cryst}$, hCldn4, mCldn19, and mCldn15 are colored blue, green, yellow, and orange, respectively. C-CPEs paired with mCldn3$_{cryst}$, hCldn4, and mCldn19 are omitted for clear display of the Cldns. Pro134 of mCldn3 is colored cyan. **b** Superposition of mCldn3$_{cryst}$ and the other structure-determined subtypes

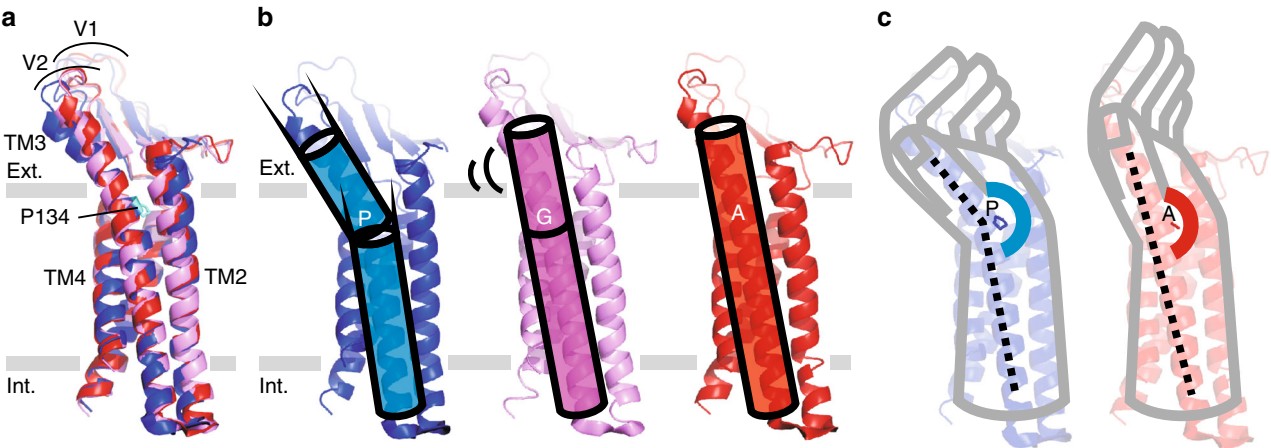

**Fig. 3** Structures of Pro134 mutants. **a** Superposition of mCldn3$_{cryst}$ (blue), mCldn3$_{cryst}$ P134G (violet), and mCldn3$_{cryst}$ P134A (red). C-CPEs paired with mCldn3$_{cryst}$, mCldn3$_{cryst}$ P134G, and mCldn3$_{cryst}$ P134A are omitted. **b** Determination of the TM3 structure with proline, glycine, or alanine. TM3 is bent in the case of proline (blue). TM3 is straight in the case of glycine, with probable swaying (violet). TM3 is straight in the case of alanine (red). **c** TM3 thenar in the left-hand model. The TM3 thenar is highlighted by the colored arc. When the TM3 thenar is proline, the bent TM3 makes the hand of the ECD lean toward the back of the hand (left). When the TM3 thenar is alanine, the straightened TM3 leads the hand of the ECD to align with the forearm of the TMD (right)

mCldn3$_{cryst}$ P134G (Supplementary Figure 2h). In a crystal lattice, the extracellular part of TM3 of chain A approaches the cytosolic part of TM1 of chain C, whereas the extracellular part of TM3 of chain C is exposed to solvent (Supplementary Figure 2g). The higher B-factor of chain C in the P134G mutant implies the flexibility of the TM3 exposed to solvent. Therefore, it is considered that the glycine on TM3 destabilizes the helix, leading to wobbling, and that the TM3 of mCldn3$_{cryst}$ and mCldn3$_{cryst}$ P134A have fixed bent and straight forms, respectively.

Considering that the residue at the TM3 thenar in the Cldn family is either proline, glycine, or alanine, the TM3 structure of each subtype could also be determined in the same manner as our three structures of mCldn3, which covered all of the TM3 types (Fig. 3b); the proline makes the TM3 helix bend as if the hand is leaning toward the back of the hand; the glycine and alanine straighten the TM3 helix as if the hand is aligned with the forearm, but the TM3 with the glycine-type probably sways.

**Influence of C-CPE binding on Cldn structure**. In addition, the structures of the two Pro134 mutants showed that the electron density of the ECH (extracellular helix) region, which was not well resolved in the case of mCldn3$_{cryst}$, could be clearly observed as a hairpin loop structure, unlike the helix of mCldn15 (Supplementary Figure 1c, d, Supplementary Figure 7a). This region has been considered to be a helical structure among Cldn subtypes, but because of the possibility that it could form a loop structure, we hereafter refer to this region as a cis-interaction nub (CIN). Previous studies of Cldn/C-CPE complex structures revealed that the C-CPE sterically hinders trans-interaction, and distorts cis-interaction. The C-CPE closely binds the β-sheet and destabilizes the CIN (ECH) structure (Supplementary Figure 4e, f), which is presumed to be a main factor in the disruption of cis-interactions;[22,23] cis-interactions are formed by hydrophobic interactions between a large hydrophobic residue on the CIN (ECH) of one molecule and the complementary hydrophobic pocket, termed the cis-interaction pocket, in the TM3-β5 region (the thumbnail in the left-hand model), of the adjacent molecule (Fig. 4c)[20]. The structures of mCldn3, however, revealed that C-CPE locates too distantly over the β-sheet to interfere with CIN (Supplementary Figure 4c, d), and the CIN structure was resolved clearly in the structures of the Pro134 mutants despite the C-CPE binding (Supplementary Figure 7a). Therefore, we focused on the

conformational change of the cis-interaction pocket as the receptor for the CIN.

The comparison of the homology model of the apo-form mCldn3$_{cryst}$ P134A, based on the mCldn15 structure, and the C-CPE-bound form mCldn3$_{cryst}$ P134A showed that C-CPE pulled the V2 loop and rotated the extracellular edge of TM3, as shown by the blue arrows in Supplementary Figure 7b, and that the residues on the upper side of the cis-interaction pocket (Phe146 and Tyr147 in mCldn3) moved parallel to the membrane plane. Similar structural changes are well conserved in the other subtypes regardless of the presence or absence of the TM3 bending (Supplementary Figure 7c). Therefore, these findings presume that disruption of the cis-interactions by C-CPE is due mainly to displacement of the upper side of the cis-interaction pocket in a direction horizontal to the membrane plane, independent of the structural change due to TM3 bending.

**Effects due to a different TM3 structure**. The angle of the TM3 helix at the extracellular side configures the overall arrangement of the extracellular β-sheet because TM3 connects through the V2 loop to β5, which composes the β-sheet together with the β1-β4 strands. The effect of the TM3 bending can be clarified by comparing the structures of mCldn3$_{cryst}$ with a bent TM3 and mCldn3$_{cryst}$ P134A with a straight TM3. The degree of TM3 bending was calculated with the TM3 thenar as the vertex, and the averaged angle of TM3 in mCldn3$_{cryst}$ (25°) and that in mCldn3$_{cryst}$ P134A (17°); hence Pro134 bends the TM3 helix by ~8° (Supplementary Figure 8a).

For a precise comparison of the structural shift resulting from the TM3 bending, the structures of mCldn3$_{cryst}$ and mCldn3$_{cryst}$ P134A were superimposed via the low root-mean-square deviation (RMSD) region in the TMD (Supplementary Figure 8b, c). The shift becomes larger in the distal regions farther from the TM3 thenar to the extracellular edge, whose structural shift reaches approximately 5 Å (Fig. 4a). The TM3 helix of mCldn3$_{cryst}$ pulls the extracellular β-sheet as a whole in the same direction as the bending. The regions largely shifted by the TM3 bending include residues involved in Cldn–Cldn interactions such as the cis-interaction pocket, V1 and V2 regions. Therefore, the TM3 bending is suggested to affect Cldn–Cldn interactions, both the cis- and trans-interaction.

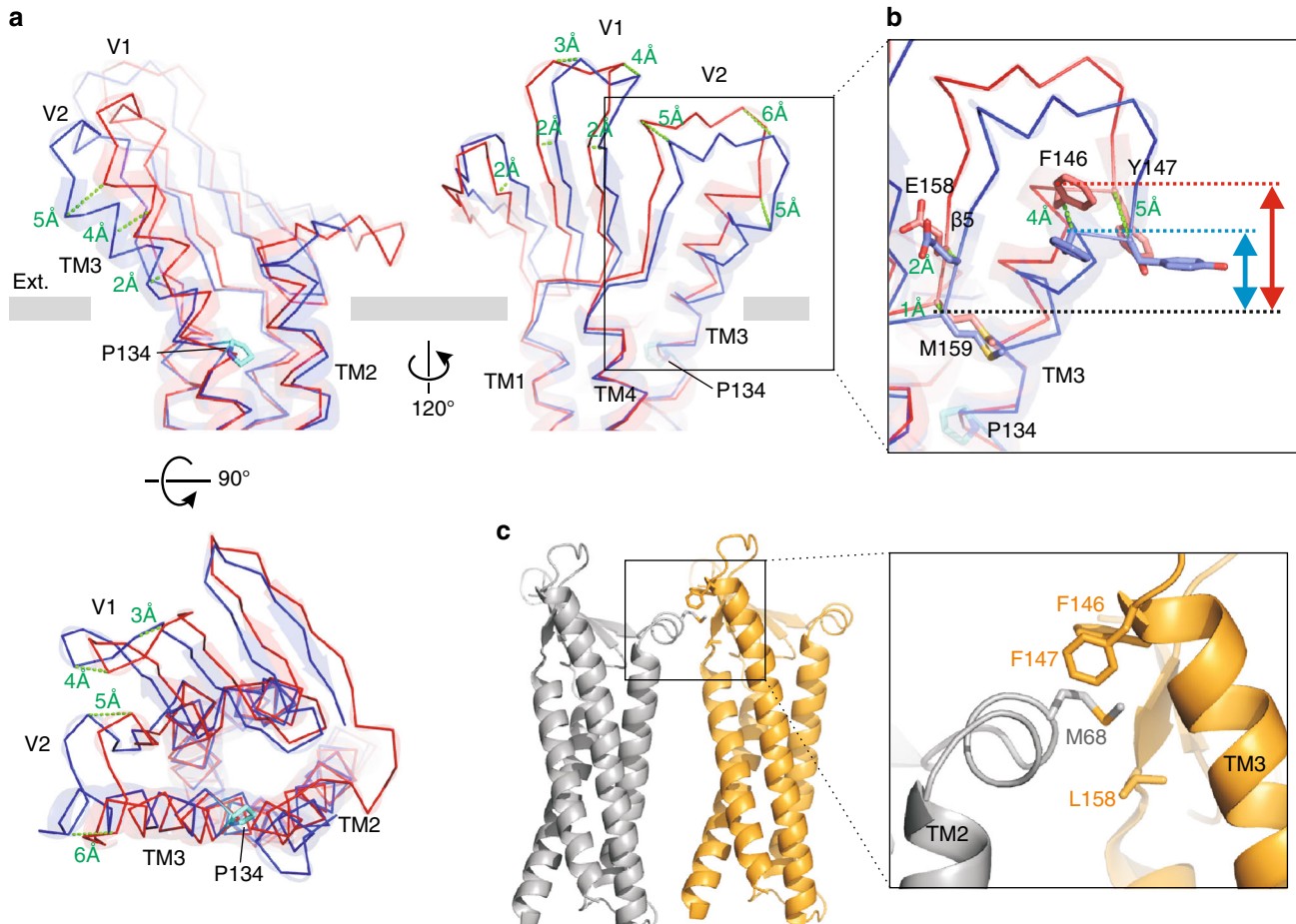

**Fig. 4** Structural shift by TM3 bending. **a** Superposition of mCldn3$_{cryst}$ and mCldn3$_{cryst}$ P134A. The Cα strands of mCldn3$_{cryst}$ and mCldn3$_{cryst}$ P134A are colored blue and red, respectively. Distances arising from the TM3 bending are shown in green. **b** Enlarged view of the cis-interaction pocket periphery. The residues constituting a cis-interaction pocket are shown in stick representation. Double-headed arrows indicate the size of the cis-interaction pocket in the direction perpendicular to the cell membrane plane. **c** Cis-interaction observed in mCldn15 crystal (PDB ID: 4p79). Two neighboring molecules of mCldn15 are shown in ribbon representation, and key residues in the cis-interaction are shown in stick representation

To evaluate how the structural shift of the ECD caused by the TM3 bending affects the TJ physiology, we examined the adhesion properties of TJs. The adhesivity of TJs was evaluated by a dissociation assay on cell sheets that were formed by stable SF7-derived cell lines, which lacked endogenous TJs, expressing the wild type or Pro134 mutant mCldn3. We confirmed that each of the wild type or mutant mCldn3 was expressed at an equivalent level among them and localized uniformly on the contact areas with the adjacent cell membranes (Supplementary Figure 9). The dissociation assay revealed that the adhesivity of TJs comprising mCldn3 P134G or P134A was significantly stronger than that of TJs comprising wild-type mCldn3 (Fig. 5a). This finding indicates that the presence or absence of TM3 bending affects the adhesion properties of TJs.

Next, to assess whether the structural shift of the ECD affects the TJ strand morphology, we observed TJ strands comprising the wild type or Pro134 mutant mCldn3 by freeze-fracture electron microscopy. Each of the three mCldn3s fused with enhanced green fluorescent protein (EGFP) was expressed on the HEK293 GnT1⁻ strain lacking endogenous TJ strands by the BacMam system[33]. The expression level and monodispersity of the proteins evaluated by fluorescence-detection size-exclusion chromatography (FSEC)[34] were comparable among the three mCldn3s (Supplementary Figure 9e). The TJ strands formed by the wild-type mCldn3 appeared almost linear with little flexibility (Fig. 5b).

On the other hand, the TJ strands formed by mCldn3 P134G or P134A were characterized as curvy and highly flexible with many hairpin curves (Fig. 5c, d). Additionally, in the case of mCldn3 P134G and P134A, some regions with highly dense TJ strands were observed, while the distribution of the strands comprised by wild-type mCldn3 was rather sparse (Fig. 5). This suggests that highly flexible strands can gather densely whereas less flexible strands tend to gather at some intervals. Therefore, these findings indicate that differences in TM3 affect TJ strand morphology, especially TJ strand flexibility and density.

Differences in TJ strand flexibility are thought to be due to differences in the Cldn–Cldn interactions, especially side-by-side cis-interaction. The main component of the cis-interaction is the hydrophobic interaction between a hydrophobic residue of CIN (ECH) and the cis-interaction pocket[20] (Fig. 4c). The residues composing the cis-interaction pocket are well conserved and aligned among Cldn family members (Supplementary Figure 5). Although a C-CPE-free cis-interacting structure of any proline-type Cldn is not available, the binding manner of C-CPE was well conserved among Cldn subtypes regardless of the residue type at the TM3 thenar. Therefore, we assume that the conformational difference at the cis-interaction pocket between proline- and alanine-type Cldns derives from the TM3 bending. The upside of the cis-interaction pocket of mCldn3 comprises Phe146 and Tyr147, which locate on the extracellular edge of TM3, and the

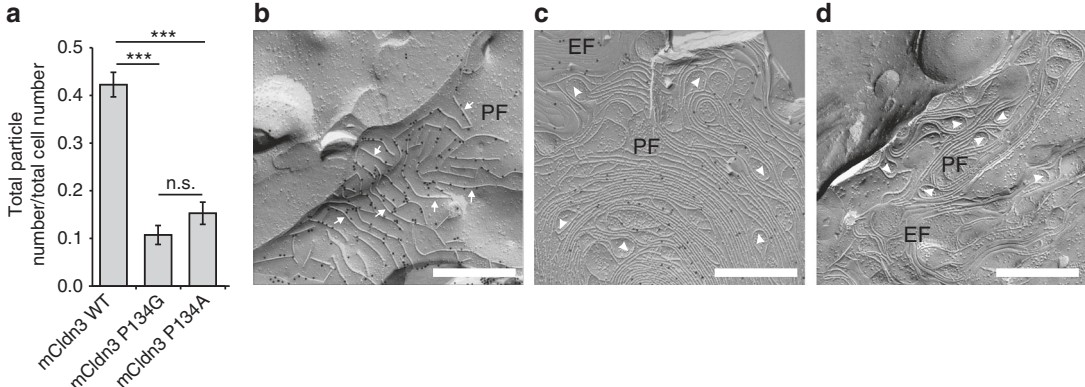

**Fig. 5** Morphology and adhesivity of TJ strands formed by mCldn3 with a different TM3 structure. **a** Adhesive property of TJs formed by a stable SF7-derived cell line expressing mCldn3. Dissociation assay showed that TJs formed by mCldn3 P134G or P134A had stronger adhesivity than TJs of mCldn3 WT ($n = 18$/group). Results from three distinct clones are shown as means ± SEM. *P*-values were calculated using a two-tailed independent *t*-test, and $P < 0.05$ was considered significant. n.s., not significant; ***$p < 0.001$. **b–d** Representative electron microscopic images of freeze-fracture replicas of TJ strands of mCldn3. **b** TJ strands consisting of wild-type mCldn3 appear as straight strands (arrow). **c** TJ strands consisting of mCldn3 P134G appear as curvy strands with many hairpin curves (arrowhead). **d** TJ strands consisting of mCldn3 P134A appear as curvy strands with many hairpin curves. PF = protoplasmic face. EF = exoplasmic face. Scale bar = 500 nm. Source data are provided as a Source Data file (**a**)

downside is framed by Glu158 and Met159 on β5, which locate next to TM3. Along with the TM3 bending, Phe146 and Tyr147 of the proline-type (mCldn3$_{cryst}$) were pushed down more toward the lipid bilayer than those of the alanine-type (mCldn3$_{cryst}$ P134A), whereas Glu158 and Met159 were hardly shifted (Fig. 4b). As a result, the vertical width between the top and bottom of the cis-interaction pocket becomes narrower due to the TM3 bending. These different sizes of the cis-interaction pocket could naturally influence the joint mobility of the hydrophobic interaction between the cis-interaction pocket and the hydrophobic residue of CIN, leading to differences in TJ strand flexibility.

**Model for TJ strand flexibility**. Based on the above comparisons, we propose a model for TJ strand flexibility that can be modified by TM3 bending (Fig. 6). The property of the residue at the TM3 thenar, which is proline, glycine, or alanine, controls the TM3 angle (Fig. 3), and then determines the size of the cis-interaction pocket; a narrower pocket in the case of a bent TM3, a wider pocket in the case of a straight TM3 (Fig. 4b). The hydrophobic residue of CIN, which fits into the pocket, would not be changed by the TM3 bending so that the difference in the pocket size would mostly influence the joint mobility of the cis-interaction. The narrow pocket would tightly grasp the hydrophobic residue of the CIN of the adjacent Cldn by sandwiching from the vertical direction, and restrict the orientation of the cis-interaction, whereas the wide pocket would loosely catch the CIN enabling wide orientation range. The permissible orientation range of cis-interaction may be determined by the relationship between the sizes of the pocket and CIN, and the TM3 structure is a major determinant of the pocket size, in addition to the kinds of residues comprising the cis-interaction pocket and CIN depending on the Cldn subtype (Supplementary Figure 5). The permissible orientation range of cis-interaction would influence TJ strand flexibility. As the result, TJ strands with little flexibility consisting of orientation-restricted cis-interactions (e.g. wild-type mCldn3) represent the linear pattern, whereas highly flexible TJ strands consisting of cis-interactions with widely permissible orientation (e.g. mCldn3 P134G or P134A) represent the curvy pattern. The FRAP assay showed that the rates and extent of fluorescence recovery are identical among wild-type mCldn3 and P134G or P134A mutant (Supplementary Figure 10). These data suggest

that despite differences in TJ strand architecture, bulk mobility of mCldn3 (and likely cis binding affinity) is independent of the TM3 bending. The flexibility of the TJ strands could affect the density of TJ strands, and a high density of TJ strands may strengthen the adhesivity of TJs, as in mCldn3 P134G and P134A (Fig. 5).

## Discussion

The three crystal structures of mCldn3 revealed that the TM3 bending structure is a subtype-specific feature. TM3 bending is determined by a single residue at the TM3 thenar, which alters the arrangement of the ECD involved in the Cldn–Cldn interactions. The long distance from the ECD to the TM3 thenar residue generates the large inclination of the ECD by the TM3 bending, affecting the Cldn–Cldn interactions that determine the morphology and physiology of the TJ strands. Our study proposes that TM3 bending alters the cis-interaction pocket size and the permissible orientation range of cis-interaction is one important factor determining TJ strand flexibility. The results recently reported by molecular dynamics simulation predicted that laterally-rotated cis-interactions are necessary for the TJ strand flexibility[35], which were similar concepts to our model for TJ strand flexibility. Other potential cis-interactions including TM3, in addition to the cis-interaction discussed above, have been suggested[32,36–38], and these would also be influenced by TM3 bending. TM3 bending greatly affected the positions of the V1 and V2 regions, which are involved in trans-interaction (Fig. 4). Because paracellular pores are formed through the trans-interactions, their size and shape would be influenced by TM3 bending, possibly affecting the barrier and permeation properties. Furthermore, it is possible that the shift of the residues involved in Cldn–Cldn interactions, a cis- or trans-interaction, is related to the selectivity of heterophilic pairing among Cldn subtypes. It has been thought that only the residues related to direct interactions determine the Cldn–Cldn interaction, but our results suggest that the residue modifying the transmembrane conformation also influences the formation and function of TJs through a global shift of the ECD structure. In the Cldn family, the residue of the TM3 thenar is either proline, glycine, or alanine, and the TM3 structure would determine the orientation of the ECD like that of our mCldn3 structures. In other members of the tetra-span-transmembrane protein family that are closely related to the

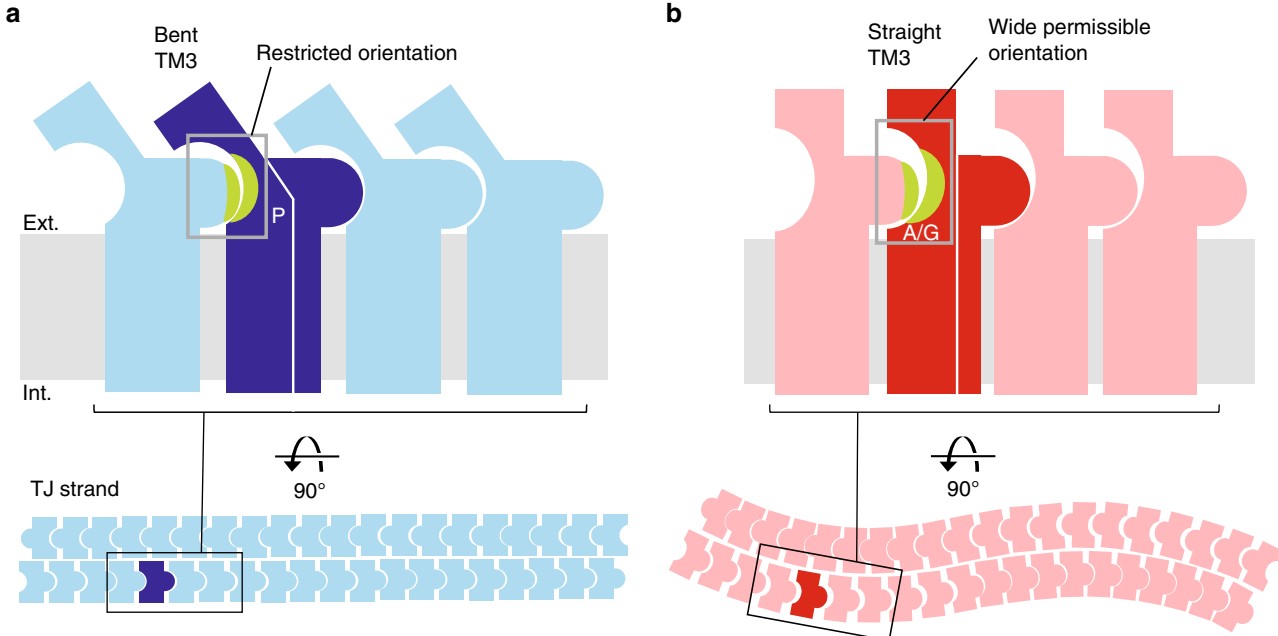

**Fig. 6** Mechanism for determining the flexibility of TJ strands by TM3 bending. **a** Proline renders TM3 bent, as seen in the mCldn3$_{cryst}$ structure, with the formation of a narrow cis-interaction pocket. The narrow pocket would restrict the orientation of the cis-interaction, resulting in linear TJ strands having little flexibility. **b** The alanine or glycine renders TM3 straight, as seen in mCldn3$_{cryst}$ P134A or P134G structure, with the formation of a wide cis-interaction pocket. The wide pocket would enable wide orientation range, resulting in curvy TJ strands with high flexibility. Hydrophobic interactions in cis-interaction are represented in green. TJ strands are depicted according to the anti-parallel double-row model

Cldn family, such as a voltage-dependent calcium channel γ1 subunit and transmembrane AMPA receptor regulatory proteins, which function as channel auxiliaries and share similar folding with Cldn[39–41], the TM3 helices are straight with serine or glycine at the TM3 thenar. This suggests that variations in the TM3 structure are a characteristic of the Cldn family, enabling generation of the optimal paracellular barriers. Further functional analyses considering the TM3 structure will facilitate our understanding of the detailed mechanisms of TJ multiformity.

## Methods

**Protein expression and purification of C-CPE.** The C-terminal fragment of CPE, residues 203–319, with alanine substituting for Ser313 (C-CPE$_{S313A}$) was subcloned into the pGEX-4T vector (GE Healthcare) and expressed with an N-terminal GST tag in Escherichia coli BL21 (DE3) (Novagene). One litter of Luria Bertani medium containing 100 µg ml$^{-1}$ ampicillin was inoculated with 10 ml of an overnight culture of BL21 (DE3) with the pGEX-4T-C-CPE$_{S313A}$. The cells were grown at 37 °C with shaking until absorbance at 600 nm reached 0.5, induced with 0.5 mM isopropyl 1-thio-β-D-galactopyranoside, and grown for another 5 h at 30 °C with shaking. The cells were harvested by centrifugation at 5000 × g for 15 min at 4 °C, and washed with phosphate buffered saline (PBS). The cell pellet was resuspended in Tris-buffered saline (TBS, 20 mM Tris-HCl, pH 7.5, 150 mM NaCl) containing 10 µg ml$^{-1}$ DNaseI, 0.5 µg ml$^{-1}$ lysozyme, and protease inhibitor cocktail (Roche), incubated for 10 min at room temperature with stirring, and disrupted by sonication on ice. The cell debris was removed by ultracentrifugation at 40,000 × g for 25 min at 4 °C. The supernatant was incubated with Glutathione Sepharose 4B (Amersham Pharmacia Biotech) for 2 h at 4 °C. The resin was loaded onto an open column and washed with TBS. The GST-tag was cleaved off with thrombin on resin overnight at 4 °C. The resin was loaded onto an open column and the tag-free C-CPE$_{S313A}$ was eluted with TBS. The eluate was concentrated and loaded onto a Superdex 200 10/300 GL column equilibrated with TBS. Fractions containing C-CPE$_{S313A}$ were concentrated to 30 mg ml$^{-1}$, flash-frozen in liquid nitrogen, and stored at −80 °C.

**Protein expression and purification of mCldn3.** The mCldn3 gene inserted in pUC vector was synthesized (GeneScript) with following modificatins; to prevent heterogeneity due to variable palmitoylation, four cysteine residues in mCldn3 (Cys103, Cys106, Cys181, and Cys182) were substituted with alanines; the last 36 residues at the C-terminus were removed. This construct, mCldn3$_{crys}$, was expressed with N-terminal tobacco etch virus (TEV) protease-cleavable 8xHis tag and a

dimerization-diminished enhanced green fluorescent protein (EGFP-mCldn3$_{crys}$). The fragment containing mCldn3$_{crys}$ gene was cleaved with BamHI and HindIII, and subcloned into the pFastBac1 vector (Life Technologies). The recombinant baculoviruses were generated according to the manufacturer's instructions. For expression, Sf9 insect cells were infected with the recombinant baculoviruses and cultured in SF900III medium (Life Technologies) for 48 h at 27 °C. The cells were collected by centrifugation at 2000 × g for 15 min, disrupted with an EmulsiFlex-C5 (Avestin) at 15,000 psi in TBS containing protease inhibitors cocktail (Roche), and the cell debris was removed by centrifugation at 2000 × g for 20 min at 4 °C. The membrane fraction was collected by ultracentrifugation at 100,000 × g for 1 h at 4 °C, resuspended in TBS using a homogenizer and solubilized for 1 h with 1%(w/v) n-dodecyl-β-D-maltopyranoside (DDM) (Anatrace) in TBS at 4 °C. Insoluble material was removed by ultracentrifugation at 100,000 × g for 45 min, and the supernatant was incubated with TALON Superflow cobalt affinity resin (Clontech) for 3 h at 4 °C. After the resin was washed with TBS containing 0.1% DDM and 10 mM imidazole, EGFP-mCldn3$_{crys}$ was eluted with TBS containing 0.1% DDM and 200 mM imidazole. The buffer of the eluate was exchanged with TBS containing 0.025% DDM by using a PD-10 desalting column (GE Healthcare). The 8xHis-EGFP tag was cleaved with TEV protease overnight at 4 °C, and removed by passing through TALON Superflow cobalt affinity resin.

The purified mCldn3$_{crys}$ was mixed with C-CPE$_{S313A}$ at a ratio of 1:1.2 (w/w) and rotated for 2 h at 4 °C. The protein complex was concentrated and loaded onto a Superdex 200 10/300 GL size-exclusion column equilibrated with 10 mM Tris-HCl, pH 7.5, 100 mM NaCl, 0.025% DDM. The fractions containing mCldn3$_{crys}$/C-CPE$_{S313A}$ complex were concentrated to 5 mg ml$^{-1}$ for crystallization.

**Crystallization.** The purified mCldn3/C-CPE complex (5 mg ml$^{-1}$) was crystallized by sitting drop vapor diffusion with a reservoir containing as follows; 0.1 M Tris-HCl pH 7.5, 0.05 M sodium chloride, 0.9 M magnesium nitrate, 20.9% PEG 2000 MME, 0.002% LMNG for mCldn3$_{cryst}$ complex; 0.1 M Tris-HCl pH 8.0, 1.2 M magnesium nitrate, 21.5% PEG 2000 MME for mCldn3$_{cryst}$ P134G complex; 0.1 M HEPES pH 7.0, 0.2 M sodium acetate, 0.9 M magnesium nitrate, 20.4% PEG 3350 for mCldn3$_{cryst}$ P134A complex. Crystals appeared in drops composed of a 1:1 (v/v) mixture of protein and reservoir solution after three days at 20 °C and grew for three weeks. Prior to data collection, the crystals were flash-frozen in liquid nitrogen after being soaked in the same mother liquor with gradually increasing concentrations of PEG 400 (from 0 to 6%).

**Structure determination.** X-ray diffraction data were collected at the SPring-8 BL41XU beamline, and processed using the XDSGUI[42]. The anisotropy of diffraction data was analyzed using the UCLA Diffraction Anisotropy Server[43]. The initial phase for mCldn3$_{cryst}$ in complex with C-CPE was determined by the

molecular replacement method with Phaser[44] using a homology model of mCldn3 and C-CPE created based on the structure of the complex of mCldn19 and C-CPE (PDB code: $3 \times 29$). The atomic model was rebuilt by manual model building in COOT[45] and refinement in PHENIX[46] and Refmac5[47] to a resolution of 3.6 Å (PDB code: 6AKE). The initial phase for mCldn3$_{cryst}$ P134A in complex with C-CPE was determined by molecular replacement with the dimer of mCldn3$_{cryst}$/C-CPE complex, and the model was manually rebuilt as well as that of mCldn3$_{cryst}$ to a resolution of 3.9 Å (PDB code: 6AKF). The initial phase of mCldn3$_{cryst}$ P134G in complex with C-CPE was determined by molecular replacement with the dimer of the mCldn3$_{cryst}$ P134A complex. Because of the resolution limit of 4.3 Å diffraction data, mCldn3$_{cryst}$ P134G structure (PDB code:6AKG) was refined by only rigid body refinement of three segment of mCldn3$_{cryst}$ P134G (residues of 1–27, 78–133, and 160–200 aa as the transmembrane domain, 28–77 aa as the extracellular domain, and 134–149 aa as the thumb of TM3 helix) and C-CPE. The model was manually rebuilt under the secondary structure restrain referring mCldn3$_{cryst}$ P134A model. The dihedral angles of peptide bond of all models were modified using Ramachandran plot calculated with MolProbity[48] (Supplementary Table 1).

**Binding assay.** The binding affinity with C-CPE was evaluated using fluorescence-detection size-exclusion chromatography (FSEC)[22,34]. The fragment containing EGFP-fused mCldn3 was subcloned into pBiEx-1 vector (Novagen) using Gibson assembly system (New England Biolabs). The site directed mutation, P134G, P134A, or L150S, was introduced in mCldn3 using primers shown in Supplementary Table 2. Sf9 cells expressing mCldn3 were resuspended in TBS containing 2% DDM, 1 mM EDTA, and protease inhibitor cocktail. The suspension was incubated with rotation for 30 min at 4 °C. The cell debris was removed by ultracentrifugation at 100,000 g for 20 min at 4 °C. The supernatant containing EGFP-mCldn3 was diluted in FSEC running buffer (20 mM Tris-HCl pH 7.5, 150 mM NaCl, 1 mM EDTA, 0.05% DDM) to ~200 pM EGFP-mCldn3, estimated by EGFP fluorescence (excitation wavelength = 488 nm, emission wavelength = 509 nm). EGFP-mCldn3 was incubated with purified GST-C-CPE at different concentrations (0–1 μM) for 2 h at 4 °C. The samples were analyzed using FSEC by detecting EGFP fluorescence. Complex formation was confirmed by a peak shift from a retention time of 4.3 min (EGFP-mCldn3) to a retention time of 3.8 min (EGFP-mCldn3/GST-C-CPE). Because mCldn3 was completely saturated at a concentration of 1 μM GST-C-CPE, the fraction of bound protein in this condition was taken as the maximum value. The binding rates were calculated by normalizing the peak height of the bounded state using that of the unbounded state, and plotted by nonlinear regression using SigmaPlot (Systat).

**Establishment of stable cell lines expressing mCldn3.** For construction of EGFP-fused mCldn3 expression plasmids, an EcoRI-XhoI fragment carrying mCldn3 wild-type (WT), P134G, or P134A genes was cloned into the EcoRI-XhoI site of the mammalian expression vector pCAGGS–EGFP downstream of EGFP. For generation of SF7 cells stably expressing EGFP-fused mCldn3, SF7 cells were co-transfected with pGK puro and pCAG-EGFP-mCldn3 (WT, P134A, or P134G) plasmids using Lipofectamine 2000 Reagent (Thermo Fisher Scientific) according to the manufacturer's instructions. Three stable clones were established after the selection with the culture medium supplemented with 1 μg ml$^{-1}$ puromycin, and used for further analyses.

**Immunostaining and immunofluorescence microscopy.** Confluent parental SF7 cells and SF7 cells stably expressing EGFP-fused mCldn3 plated on glass coverslips were cultured for three days, and then fixed with ice-cold methanol for 5 min. After three washes with PBS, the samples were blocked with PBS containing 1% BSA at room temperature for 10 min. Then, the samples were stained with a rabbit anti-GFP antibody (Invitrogen; Cat. No. A-6455; dilution ratio, 1:500) and rat anti-ZO1 antibody (eBioscience; Cat. No. 14-9776-80; dilution ratio, 1:1000) for 1 h at room temperature. After three washes with PBS, the samples were stained with an Alexa Fluor 488-conjugated secondary antibody (Jackson ImmunoResearch Laboratories; Cat. No. 711-545-152; dilution ratio, 1:1000) and Alexa Fluor 568-conjugated secondary antibody (Molecular Probes; Cat. No. A11077; dilution ratio, 1:1000) for 1 h at room temperature. After washing three times with PBS and once with deionized distilled water, the samples were embedded in the fluorescence mounting medium (Dako Japan). Immunofluorescence images were acquired by a confocal laser scanning microscopy LSM710 (Carl Zeiss Japan) equipped with a 488-nm argon laser and 561-nm DPSS lase as well as with C-Apochromat 40 × /1.2 W Corr M27 objective lens. Acquired images were analyzed by ZEN 2012 (Carl Zeiss Japan).

**Western blot.** To prepare the total cell lysate of SF7 cells and SF7 cells stably expressing EGFP-fused mCldn3, cells were washed two times with the solution (120 mM NaCl, 10 mM NaHCO$_3$, 5 mM KCl, 1.2 mM CaCl$_2$, 1.0 mM MgCl$_2$, 10 mM Tris-HEPES, pH 7.4), and lysed with sodium dodecyl sulfate-polyacrylamide gel electrophoresis (SDS-PAGE) sample buffer [62.5 mM Tris-HCl (pH 6.8), 2% SDS (w/v), 5% 2-mercaptoethanol (v/v), 10% glycerol (v/v), and 0.025% bromophenol blue (w/v)], sonicated, and boiled for 2 min at 95 °C. Equal amounts of protein samples were separated by SDS-PAGE, and transferred onto a polyvinylidene difluoride membrane. The membrane was washed two times with TBS containing 0.1% Tween-20 (v/v) (TBST), and blocked with TBST containing 5% skim milk (w/v) at room temperature for 30 min. After blocking, the membrane was blotted with a rabbit anti-Cldn3 antibody (Life Technologies; Cat. No. 34-1700; dilution ratio, 1:2000) or mouse anti-β-Actin (Sigma-Aldrich Japan; Cat. No. A5441; dilution ratio, 1:2000) for 1 h at room temperature. After washing three times with TBST for 10 min, the membrane was incubated with an HRP-labeled anti-rabbit antibody (GE Healthcare; Cat. No. NA934; dilution ratio, 1:3000) for 1 h at room temperature. After washing three times with TBST for 10 min, the membrane was reacted with Immobilon Western Chemiluminescent HRP Substrate (Merck Millipore Japan), and chemiluminescent signal was acquired by Image Quant LAS 4000 (GE Healthcare).

**Dissociation assay.** Confluent parent SF7 cells and SF7 cells stably expressing EGFP-fused mCldn3 on on cell culture dishes with a 3.5-cm diameter were cultured for three days. Dissociation assays were performed according to the reported procedure[49] as follows. After removal of the medium, 1 ml of the dissociation buffer (150 mM NaCl, 2 mM CaCl$_2$, 10 mM HEPES, pH 7.4) was gently added, and cell sheets were gently scraped with the cell scraper (Sumitomo Bakelite Co., Ltd.). The cell sheets in the buffer were repeatedly pipetted (10 times in Fig. 5a, or 8 times in Supplementary Figure 9d) with bovine serum albumin-coated pipette tips, and fixed by the addition of 100 μl of 25% glutaraldehyde solution. Then, the extent of cell dissociation was represented by the ratio of the total particle number to the total cell number.

**Freeze-fracture electron microscopy.** For transfection of mCldn3 into mammalian cells, baculoviruses expressing EGFP-mCldn3 were generated using the Bac-Mam system[33]. For expression, HEK293 GnT1$^-$ cells were infected with the BacMam recombinant baculoviruses and cultured with adhesion in Pro293a medium (Lonza) supplemented with 4% fetal bovine serum, GlutaMAX (Thermo Fisher), PenStrep (Gibco), and 2 mM sodium butyrate for 48 h at 37 °C. The cells were fixed using 0.1 M phosphate buffer containing 4% formaldehyde for 1 h at 4 °C; cryo-protected by overnight incubation with PBS containing 30% glycerol, and 0.01% sodium azide at 4 °C; and snap frozen by immersion into liquid nitrogen. Replica membrane was prepared using JFD-II freeze-etching system (JEOL). The frozen cells were fractured with a metal knife cooled to −150 °C, and shadowed by unidirectional platinum-carbon evaporation from a 60° angle, followed by rotary carbon evaporation from the top. The cell debris was digested in a solution containing 2.5% SDS, 20% sucrose, and 15 mM Tris-HCl pH 8.3 for 20 min at 120 °C. The replica membrane was immunostained using anti-GFP antibody (Abcam; Cat. No. ab6556; dilution ratio, 1:200) and then anti-rabbit antibody conjugated to 15-nm colloidal gold (BBI Solutions; Cat. No. CRL-1011-50; dilution ratio, 1:200). The replicas were imaged in JEM-1010 electron microscopy (JEOL) equipped with a 4k × 4k CMOS camera TemCam F416 (TVIPS).

**FRAP analysis.** Confluent SF7 cells stably expressing EGFP-fused mCldn3 on glass bottom dishes with a 3.5-cm diameter (AGC TECHNO GLASS CO., LTD.) were cultured for 3 days. FRAP analysis was performed according to the reported procedure[50] as follows. The cells were pre-incubated in FluoroBrite DMEM Media (Thermo Fisher Scientific K.K.) with 10% fetal bovine serum, 2 mM L-glutamine, and 10 μM forskolin for 6 h, and processed for the FRAP analysis. FRAP analysis was performed by the LSM880 confocal laser scanning microscopy with the Plan-Apochromat 63 × 1.4 Oil DIC M27 objective lens (Carl Zeiss Japan). A heated chamber was used to keep the temperature at 37 °C and maintain the CO$_2$ in the chamber at 5%. The images were collected at 1 frame/sec with the following parameters: resolution, 512 × 512 pixels; pinhole, 1.43 airy unit; excitation wavelength, 488 nm; and laser transmission, 1.0%. A targeted region was bleached using 100% laser transmission at 488 nm (1 pulse). The fluorescence intensity was normalized to the prebleach intensity using ImageJ software (National Institutes of Health, Bethesda, MD).

## Data availability

Coordinates and structure factors of mCldn3$_{cryst}$, mCldn3$_{cryst}$ P134A, and mCldn3$_{cryst}$ P134G have been deposited in the Protein Data Bank under accession codes 6AKE, 6AKF and 6AKG, respectively. The source data underlying Fig. 5a and Supplementary Figs 6a, 9b, 9c, 9d and 10b are provided as a Source Data file. Other data are available from the corresponding author upon reasonable request.

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

## Acknowledgements

This work was supported by Grants-in-Aid for Scientific Research (S), the Japan Agency for Medical Research and Development (AMED), Grants-in-Aid for Scientific Research (A), and Core Research for Evolutional Science and Technology (CREST). The synchrotron radiation experiments were performed at BL41XU in SPring-8 with the approval of the Japan Synchrotron Radiation Research Institute (JASRI Proposal numbers: 2015A1090, 2015B1042, 2016A2697, and 2016B2721) and at BL2S1 at the Aichi Synchrotron Radiation center (Proposal No: 2015N2004, 2015N3003, 2015N6006, 2016N3006, and 2016N4007). We thank the beamline staff for excellent facilities and support.

## Author contributions

S.N. performed protein expression, purification, crystallization, and binding assay, and collected and processed diffraction data. K.I. solved, refined, and analyzed the structure. H.T. collected fluorescence microscopy images, and performed dissociation assay and FRAP analysis. S.N. and K.N. took electron microscopy images. H.S. and Y.S. screened claudin genes. S.N., K.I. and Y.F. wrote the manuscript, and all authors commented on the paper. A.T., S.T. and Y.F. supervised the research.

## Additional information

**Competing interests:** The authors declare no competing interests.

