## [Peer Review File · Nature Communications]

Reviewers' comments:

Reviewer #1 (Remarks to the Author):

In the manuscript NCOMMS-18-22628-T Fujiyoshi and colleagues present the crystal structure of claudin-3 in complex with C-CPE S313A at 3.4 Å resolution. The structural analysis of mutants in TM3 of claudin-3 revealed that this helix is either straight or bent depending on a proline residue. This indicated that TM3 bending is a claudin subtype-specific feature. The bending does not affect the mode or affinity of binding of C-CPE to claudins. However, the position of C-CPE relative to the beta-sheet and the "ECH/CIN" in the ECD of claudins is changed. Based on the data the authors propose (i) that C-CPE inhibits cis-interaction of claudins by affecting the "cis-interaction pocket" and thereby actively disrupts tight junction strands (ii) that TM3 bending affects cis-interaction mediated by "ECH/CIN" and the "cis-interacting pocket" and thereby influences the lateral flexibility of claudin strands. Experimentally, this is a very solid study which provides novel structural data about claudins and their interaction with C-CPE. Moreover, the influence by Pro and Gly or Ala on the bent and respective straight conformation of TM3 is clearly demonstrated by crystal structures. However, major claims (i, ii) are not convincingly justified by the data presented. I have the following major comments:

1) Authors: li 93 "C-CPE binds to two loops of mCldn3 – the loop between beta1 and beta2 (V1 region) and the loop between TM3 and beta5 (V2 region), in which residues related to C-CPE binding are at the same position as those of mCldn19 and hCldn4"

Reviewer: Unfortunately, structural comparison of claudin residues directly involved in C-CPE binding to explain the different affinities of Cldn3 and -4 vs Cldn19 and C-CPE-wt vs C-CPE-S313A is missing. Which structural features/interactions cause the different affinity of Cldn3/-4 vs Cldn19 and C-CPEwt vs C-CPE-S313A? Why is this not discussed in detail, for instance, like in Shinoda et al., 2016, Sci Rep. for the Cldn4:C-CPE-wt / Cldn19:C-CPE-S313A comparison?

2) Authors: li 115 "The TM3 bending ... have not been discussed in previous studies"

Reviewer: The claudin TM3 bending and its influence on C-CPE binding has already been discussed in Shinoda et al., Sci Rep, 2016. In addition, TM3 bending of claudin-3/-5 and the influence of P134/135 and other residues on TM3 bending has been investigated in Irudayanathan et al., Ann N Y Acad Sci., 2017, 1405:131-146. and Rossa et al. Biochem J. 2014 464:49-60 as well. Please discuss your findings in relation to the mentioned published data.

3) Authors: li 167 "Therefore, these findings indicate that disruption of the cis-interactions by C-CPE is due mainly to displacement of the upper side of the cis-interaction pocket in a direction horizontal to the membrane plane"

Reviewer: Where is it shown that C-CPE binding disrupts cis-interactions directly? There is enough evidence that the "CIN" and the "cis-interaction pocket" are involved in the interaction between claudin protomers. However, even if - as proposed - these two regions interact directly in native TJ strands, it is still questionable if the proposed conformational change of these regions upon C-CPE binding are causative for TJ disruption. What is the evidence that disruption of the putative "CIN/pocket" cis-interaction but not steric prevention of trans-interaction is the main cause of disassembly of TJ strands? Please explain/discuss.

4) Authors: li 184 "The regions largely shifted by the TM3 bending include residues involved in Cldn-Cldn interactions such as the cis-interaction pocket, V1 and V2 regions. Therefore, the TM3 bending is suggested to affect Cldn-Cldn interactions, both the cis- and trans-interaction."

Reviewer: Due to the proposed large shift of interaction-relevant regions Cldn-Cldn interaction could be affected also outside the "cis-interaction pocket". Please discuss.

5) Authors: li 203 "The TJ strands formed by the wild-type mCldn3 appeared almost linear with little flexibility (Fig. 4b). On the other hand, the TJ strands formed by mCldn3 P134G or P134A were characterized as curvy and highly flexible with many hairpin curves (Fig. 4c, d). Additionally, in the case of mCldn3 P134G and P134A, some regions with highly dense TJ strands were observed, while the distribution of the strands comprised by wild-type mCldn3 was rather sparse (Fig. 4). This suggests that highly flexible strands can gather densely whereas less flexible strands tend to gather at some intervals. Therefore, these findings indicate that differences in TM3 affect TJ strand morphology, especially TJ strand flexibility and density."

Reviewer: Previous studies indicated that Cldn3 forms rather curved than straight strands (Furuse et al., *J Cell Biol.* 1999, 147:891-903; Milatz et al., *Biochim Biophys Acta.* 2010, 1798:2048-57, Rossa et al., *J Biol Chem.* 2014 289:7641-53). Hence, the data shown here are not fully convincing and inconsistent with previous reports. Morphometric quantification is essential to underline the data. Can you exclude that the curved strands found for the Cldn3 mutants are a secondary consequence, for instance, of a high strand density?

6) Authors: li 212 "The main component of the cis-interaction is the hydrophobic interaction between a hydrophobic residue of CIN (ECH) and the cis-interaction pocket"

Reviewer: It should be mentioned that proposal of this interaction is based on the crystal packing of claudin 15 (Suzuki et al., 2014).

9) Model for TJ strand flexibility

Reviewer: The proposed model is one conceivable explanation of the striking flexibility of most TJ strands. However, the model remains speculative, e.g. since the morphological data are not conclusive. Please consider the recent study of Zhao et al., *Communications Biology*, 2018 1, 50 which focused on structural determinants of TJ strand flexibility.

10) Authors: li 256 "It has been thought that only the residues related to direct interactions determine the Cldn-Cldn interaction, but our results suggest that the residue modifying the transmembrane conformation also influences the formation and function of TJs through a global shift of the ECD structure"

Reviewer: It is rather a long standing problem to distinguish direct and indirect influences of residues on the interaction between claudin protomers. Other potential cis-interfaces including TM3 have been suggested (Gong et al., *Mol Biol Cell.* 2015 26:4333-46; Irudayanathan et al., *J Phys Chem B.* 2016 120:77-88 Irudayanathan et al., *Ann N Y Acad Sci.*, 2017, 1405:131-146, Piontek et al., *Ann N Y Acad Sci.*, 2017 1397:143-156). Hence, TM3 bending could influence TJ formation independent of the "CIN/pocket" interaction discussed in this study. Please consider these in the discussion.

Minor comments:

11) Authors: li 113 "While the structures of mCldn15 and mCldn19, which possess alanine at this position, do not show TM3 bending, the TM3 of hCldn4 with proline is certainly bent"

Reviewer: At the "TM3 thenar", Cldn3 and Cldn4 but not Cldn19 contain a proline. Similarly, TM3 of Cldn3 and Cldn4 but not that of Cldn19 are mentioned to be bent. However, it is mentioned that not Cldn3 and Cldn4 but Cldn19 and Cldn4 are positioned similar to the "palm". This seems to be contradictory. Please discuss/explain.

Reviewer #2 (Remarks to the Author):

Claudin-family proteins provide the structural basis for the control of paracellular permeability, which is specifically dependent on differential claudin expression. To date only a handful of claudins have been structurally characterized at high resolution. In this manuscript, the authors solved a novel structure for claudin-3 at 3.4 Angstrom resolution. The major finding of the study was that the third transmembrane domain (TM3) of claudin-3 has a key proline residue (P134) that results in a bend that is not present in other claudins harboring a glycine or alanine residue (e.g. claudin-15 and -19). P134G and P134A mutants of claudin-3 caused the TM3 domain to be straight and comparable to claudin-15. The structure determinations were well controlled for potential confounding factors, such as the requirement for CPE to get crystals to form as well as antiparallel interactions between transmembrane domains in the crystals.

Freeze fracture studies of transfected claudin-null cells demonstrated that wt claudin-3 produced tight junction strands at a lower density than those of claudin-3 P134G or P134A, which is a significant observation. A model is proposed that bent vs straight TM3 containing claudins have different cis-packing that leads to altered strand density. However, the model would benefit from additional testing and verification.

Specific comments

As the authors point out there are several other claudins that have a comparable proline residue - most notably claudin-4 for which the structural of the CPR-claudin-4 complex has also been solved at 3.5 Angstrom resolution. However, in the comparison in Figure 1c it looks as though the bend is different for claudin-4 vs claudin-3. Is this the case? If so, then the authors need to explain why this is the case

Also, it is difficult to visualize Figure 1c. Presenting claudin-3, -4, -15 and -19 as individual molecules to highlight bent vs straight TM3 in addition to the the overlay would be extremely helpful (similar to what was done in Extended Figure 4)

p1 "TJ strands among different epithelia have different morphologies, and the TJ strand morphology such as number of strands is suggested to correlate with paracellular permeability". In fact, this is not the case, as demonstrated by Colegio, et al., 2003. *Am. J. Physiol. Cell Physiol.* 284:C1346-54. Also, in the Colegio study, claudin-2 and claudin-4 were found to have dramatically different strand density even though both have the TM3 proline residue comparable to claudin-3. A more thorough literature survey of claudin-specific tight junction strand architecture would strengthen the manuscript as would explaining how TM3 bends in different claudins can lead to different strand architecture.

The adhesive assay using transfected SF7 cells is somewhat informative, but is highly sensitive to parameters such as the pipetting frequency (e.g. Figure 4a vs Extended Figure 9d). If trans-interactions are different for wild type vs P134G and P134A claudin-3, this should be detectable by a difference in barrier function (e.g. TER or flux of small molecules) that could be done with transfected mammalian cells. Wild type claudin-3 would also be predicted to have a higher rate of turnover than P134G or P134A claudin-3, which is also testable.

The model presented in Figure 5 and described in lines 230-246 was not directly tested. If the change in cis interactions occurs when proline is mutated to glycine or alanine, then differences in mobility between wild type and point mutant claudin-3 could be directly detected by a technique such as FRAP. This is feasible considering that pre-existing EGFP-tagged claudin-3 constructs have already been developed and would strengthen the study.

Reviewer #3 (Remarks to the Author):

The structures presented in this work appear to be properly determined and refined. I have a few suggestions that should be addressed.

Given the low resolution of the data can the authors comment on what restraints were used in the refinement in the methods section.

As noted in the header of the PDB validation reports that were provided, the report does not correspond to a PDB submitted structure. There is no PDB code listed in the manuscript for the three structures determined in the paper. The authors should complete the PDB submission and provide the official validation report for the PDB processed structures along with a PDB code for each structure. The paper should not be published until this is completed.

comments on Extended Table 1:

The authors should report the appropriate number of significant figures for the Rmerge, Rpim, Mean I(s)I, Completeness and average B factors. For example, a RMerge value of 0.05188 is too many digits for the level of precision. See commentary on appropriate reporting of values (Dauter & Baker, *Acta Cryst D*, 2007 Mar; 63(Pt 3):275). The authors should go through the table and correct these.

The Rmerge value for the highest resolution data shell is quite high in all three structures. The authors should provide the CC1/2 and CC* values for the highest resolution shell to assess whether these reflections are useful in refinements and what the resolution cutoff of the data should be.

No. of atoms and average B-factors should be reported separately for each chain in the structure.

Given the low resolution of the data, the authors should provide a Ramachandran plot with distribution of residues and a Molprobity score to assess the quality of the geometry of the structures.

Reviewers' comments:

Reviewer #1 (Remarks to the Author):

In the manuscript NCOMMS-18-22628-T Fujiyoshi and colleagues present the crystal structure of claudin-3 in complex with C-CPE S313A at 3.4 Å resolution. The structural analysis of mutants in TM3 of claudin-3 revealed that this helix is either straight or bent depending on a proline residue. This indicated that TM3 bending is a claudin subtype-specific feature. The bending does not affect the mode or affinity of binding of C-CPE to claudins. However, the position of C-CPE relative to the beta-sheet and the “ECH/CIN” in the ECD of claudins is changed. Based on the data the authors propose (i) that C-CPE inhibits cis-interaction of claudins by affecting the “cis-interaction pocket” and thereby actively disrupts tight junction strands (ii) that TM3 bending affects cis-interaction mediated by “ECH/CIN” and the “cis-interacting pocket” and thereby influences the lateral flexibility of claudin strands. Experimentally, this is a very solid study which provides novel structural data about claudins and their interaction with C-CPE. Moreover, the influence by Pro and Gly or Ala on the bent and respective straight conformation of TM3 is clearly demonstrated by crystal structures. However, major claims (i, ii) are not convincingly justified by the data presented. I have the following major comments:

1)

Authors: li 93 “C-CPE binds to two loops of mCldn3 – the loop between beta1 and beta2 (V1 region) and the loop between TM3 and beta5 (V2 region), in which residues related to C-CPE binding are at the same position as those of mCldn19 and hCldn4”

Reviewer: Unfortunately, structural comparison of claudin residues directly involved in C-CPE binding to explain the different affinities of Cldn3 and -4 vs Cldn19 and C-CPE-wt vs C-CPE-S313A is missing. Which structural features/interactions cause the different affinity of Cldn3/-4 vs Cldn19 and C-CPEwt vs C-CPE-S313A? Why is this not discussed in detail, for instance, like in Shinoda et al., 2016, Sci Rep. for the Cldn4:C-CPE-wt / Cldn19:C-CPE-S313A comparison?

Response)

We recognize the importance of identifying the structural features related to C-CPE affinity based on the structure of mCldn3, and are proceeding with mutant experiments. We planned to discuss the determinants of C-CPE affinity in a separate paper because we already have too many figures in this manuscript. Therefore, we did not discuss it in detail in this paper. Our results revealed that the difference in C-CPE affinity between Cldn3/4 and Cldn19 depends on the residue at the L150 position in mCldn3. Substitution of the leucine with serine, the corresponding residue in Cldn19, markedly decreased the affinity (see figure below). Conversely, Saitoh et al. (Science, 2015) reported that leucine substitution of Ser152 in mCldn19 increases the affinity. In the mCldn3 structure, the side-chain of the leucine interacts with the hydrophobic pocket of the C-CPE formed by the tyrosine cluster (see figure below). Therefore, the properties of the side-chain in this position would determine C-CPE affinity.

As the reviewer suggested, this finding is important to our study, so we would like to add the figure shown below to Extended data Fig. 6 and the following sentence (line 101): “Mutational experiments revealed that substitution of Leu150 in mCldn3 for serine, the corresponding residue in Cldn19, markedly decreased C-CPE affinity (Extended

Data Fig. 6). Conversely, leucine substitution of Ser152 in mCldn19 increased the affinity²². A side-chain of this residue protrudes into a hydrophobic pocket comprising tyrosines of the C-CPE, and the interaction in this position determines C-CPE affinity.” (Ref. 22: Saitoh et al., Science. 2015, 347, 775-778)

On the other hand, our mutational experiments in the previous paper (Nakamura et al., 2018, Acta Crystallogr. F Struct. Biol. Commun. 74, 150-155) indicated that the affinity of C-CPE S313A vs wild-type was almost equivalent (see figure below), while Shinoda and colleagues speculated based on structural information alone that the S313A mutation greatly influences the affinity. We used the S313A mutant because this mutation enhances the thermostability of the Cldn/C-CPE complex and improves the resolution of the crystal, but the binding affinity of the S313A mutant did not differ significantly from that of the wild-type.

Fig.

(a) Dose-response curves of mCldn3 WT and mutants. (b) A key residue for C-CPE affinity. mCldn3 and C-CPE are colored blue and magenta, respectively.

Fig. Binding affinity of C-CPE_{WT} and C-CPE_{S313A} for mCldn3.

Apparent $K_{0.5}$ values of C-CPE_{WT} and C-CPE_{S313A} are 2.5 and 0.9 nM, respectively. (Nakamura et al., 2018, Acta Crystallogr. F Struct. Biol. Commun. 74, 150-155)

2)

Authors: li 115 “The TM3 bending ... have not been discussed in previous studies”

Reviewer: The claudin TM3 bending and its influence on C-CPE binding has already been discussed in Shinoda et al., Sci Rep, 2016. In addition, TM3 bending of claudin-3/-5 and the influence of P134/135 and other residues on TM3 bending has been investigated in Irudayanathan et al., Ann N Y Acad Sci., 2017, 1405:131-146. and Rossa et al. Biochem J. 2014 464:49-60 as well. Please discuss your findings in relation to the mentioned published data.

Response)

We apologize for our miscommunication. We intended to emphasize that we examined the cause and effect of TM3 bending for the first time and revealed that TM3 bending derives from only one residue (Pro134) in the middle of TM3. Indeed, Shinoda and colleagues also recognized the TM3 bending, but their interpretation was that it was caused by C-CPE binding and they did not mention the effect of proline. Although Irudayanathan and Rossa simulated a proline-kink in TM3 from primary sequence analysis, the actual structure and its effects were not discussed. Our structures demonstrated for the first time that the bending is caused by the proline and how it influences the overall structure of the extracellular domain and TJ strand formation. We further evaluated TM3 bending and suggested that there are only three variations of TM3 structures depending on the Cldn subtype.

To avoid misleading the readers, we changed the sentence and provided additional references in this part, as follows: “The TM3 bending was predicted and recognized in previous studies^{23,30,31}, but its cause and effect were not clarified or comprehensively discussed.” (Ref.23: Rossa et al. Biochem J. 2014, 464, 49-60, Ref.30: Shinoda et al., Sci Rep, 2016, 6, 33632, Ref.31: Irudayanathan et al., Ann N Y Acad Sci., 2017, 1405, 131-146)

3)

Authors: li 167 “Therefore, these findings indicate that disruption of the cis-interactions by C-CPE is due mainly to displacement of the upper side of the cis-interaction pocket in a direction horizontal to the membrane plane”

Reviewer: Where is it shown that C-CPE binding disrupts cis-interactions directly? There is enough evidence that the “CIN” and the “cis-interaction pocket” are involved in the interaction between claudin protomers. However, even if - as proposed - these two regions interact directly in native TJ strands, it is still questionable if the proposed conformational change of these regions upon C-CPE binding are causative for TJ disruption. What is the evidence that disruption of the putative “CIN/pocket” cis-interaction but not steric prevention of trans-interaction is the main cause of disassembly of TJ strands ? Please explain/discuss.

Response)

We did recognize that inhibition of trans-interaction by steric hindrance of C-CPE is important as a major cause of TJ disruption. At the same time, based on the structural information obtained from mCldn3 and other Cldn subtypes, the cis-interaction is also affected by C-CPE binding. In the section “Influence of C-CPE binding on Cldn structure”, we focused on the effects of C-CPE binding on the cis-interaction as another cause of TJ disruption, and discussed new findings that previous structural information did not provide.

To avoid misleading readers into concluding that the C-CPE does not affect trans-interaction, we changed the sentences at line 150 as follows: “Previous studies of Cldn/C-CPE complex structures revealed that the C-CPE

sterically hinders trans-interaction, and distorts cis-interaction. The C-CPE closely binds the β -sheet and destabilizes the CIN (ECH) structure (Extended Data Fig. 4e, f), which is presumed to be a main factor in the disruption of cis-interactions.”

4)

Authors: li 184 “The regions largely shifted by the TM3 bending include residues involved in Cldn-Cldn interactions such as the cis-interaction pocket, V1 and V2 regions. Therefore, the TM3 bending is suggested to affect Cldn-Cldn interactions, both the cis- and trans-interaction.”

Reviewer: Due to the proposed large shift of interaction-relevant regions Cldn-Cldn interaction could be affected also outside the “cis-interaction pocket”. Please discuss.

Response)

TM3 bending greatly shifted the positions of the V1 and V2 regions, which are involved in trans-interactions (Fig. 3). Therefore, it is presumed that TM3 bending influences the size and shape of paracellular pores that are formed through trans-interactions, and thus the barrier and permeation properties of TJ. We hesitated to describe this possibility, however, because structural information on trans-interaction is not available and it is not known how exactly TM3 bending influences the trans-interaction mode. Further studies are needed to analyze the detailed structure of trans-interaction and the functional effects on permeability, but following the reviewer’s suggestion, we are happy to mention it in the discussion section as follows: “TM3 bending greatly affected the positions of the V1 and V2 regions, which are involved in trans-interaction. Because paracellular pores are formed through the trans-interactions, their size and shape would be influenced by TM3 bending, possibly affecting the barrier and permeation properties.”

5)

Authors: li 203 “The TJ strands formed by the wild-type mCldn3 appeared almost linear with little flexibility (Fig. 4b). On the other hand, the TJ strands formed by mCldn3 P134G or P134A were characterized as curvy and highly flexible with many hairpin curves (Fig. 4c, d). Additionally, in the case of mCldn3 P134G and P134A, some regions with highly dense TJ strands were observed, while the distribution of the strands comprised by wild-type mCldn3 was rather sparse (Fig. 4). This suggests that highly flexible strands can gather densely whereas less flexible strands tend to gather at some intervals. Therefore, these findings indicate that differences in TM3 affect TJ strand morphology, especially TJ strand flexibility and density.”

Reviewer: Previous studies indicated that Cldn3 forms rather curved than straight strands (Furuse et al., J Cell Biol. 1999, 147:891-903; Milatz et al., Biochim Biophys Acta. 2010, 1798:2048-57, Rossa et al., J Biol Chem. 2014 289:7641-53). Hence, the data shown here are not fully convincing and inconsistent with previous reports. Morphometric quantification is essential to underline the data. Can you exclude that the curved strands found for the Cldn3 mutants are a secondary consequence, for instance, of a high strand density?

Response)

We are aware of cases that differ from the strand morphology reported in our paper. It is also reported, however, that the morphology of TJ strands may change even for the same Cldn subtype depending on the expression systems (Milatz et al. Ann. N. Y. Acad. Sci. 2017, 1405, 102-115). We believe that the BacMam expression system used in this

study has the advantage of decreasing the effects of endogenous TJ-related factors, because the transgene can be expressed efficiently and at a high level. The same expression system was used for the wild-type and two mutants, and all replicas were prepared in the same manner. We observed TJ strands in many replicas, and even at relatively low-density places, the characteristics of the TJ strands in the wild-type were still linear whereas those of the mutants were curved (see figures below). Therefore, the possibility of a secondary consequence due to strand density can be excluded, and the difference in the strand morphology can be assumed to reflect the properties of the proteins, i.e., the effect of only the presence or absence of TM3 bending at residue 134.

6)

Authors: li 212 “The main component of the cis-interaction is the hydrophobic interaction between a hydrophobic residue of CIN (ECH) and the cis-interaction pocket”

Reviewer: It should be mentioned that proposal of this interaction is based on the crystal packing of claudin 15 (Suzuki et al., 2014).

Response)

We apologize that we did not include the reference for this sentence, and appreciate your suggestion. We now cite the above-mentioned paper for this sentence.

9)

Model for TJ strand flexibility

Reviewer: The proposed model is one conceivable explanation of the striking flexibility of most TJ strands.

However, the model remains speculative, e.g. since the morphological data are not conclusive. Please consider the recent study of Zhao et al., Communications Biology, 2018 1, 50 which focused on structural determinants of TJ strand flexibility.

Response)

Zhao and colleagues argued that the cis-interface found by computational methods contributes to the flexibility of TJ

strands. They attempted to evaluate three candidate cis-interaction dimers of Cldn15, and claimed that one of them, whose interaction mode is more similar to that found in the mCldn15 crystal than the other candidates, was the most favorable candidate. However, they hypothesized that the interaction between CIN and the cis-interaction pocket is not important for the cis-interaction. In contrast, we confirmed that residues at CIN in several other subtypes are essential for TJ strand formation (data not shown). Furthermore, they did not consider a cross-linking experiment to evaluate the anti-parallel double strand (Suzuki et al., J. Mol. Biol. 2015, 427, 291-297). Their results were not consistent with the results of the cross-linking experiment. Therefore, we do not think that it is appropriate to refer to their arguments in our paper.

10)

Authors: li 256 “It has been thought that only the residues related to direct interactions determine the Cldn-Cldn interaction, but our results suggest that the residue modifying the transmembrane conformation also influences the formation and function of TJs through a global shift of the ECD structure”

Reviewer: It is rather a long standing problem to distinguish direct and indirect influences of residues on the interaction between claudin protomers. Other potential cis-interfaces including TM3 have been suggested (Gong et al., Mol Biol Cell. 2015 26:4333-46; Irudayanathan et al., J Phys Chem B. 2016 120:77-88 Irudayanathan et al., Ann N Y Acad Sci., 2017, 1405:131-146, Piontek et al., Ann N Y Acad Sci., 2017 1397:143-156). Hence, TM3 bending could influence TJ formation independent of the “CIN/pocket” interaction discussed in this study. Please consider these in the discussion.

Response)

Thank you for your suggestion. The possibility of other cis-interfaces must be considered, even if we regard the cis-interface observed in mCldn15 crystal as the most conceivable. The shift induced by TM3 bending, as revealed in this paper, could also influence these potential cis-interfaces. Therefore, we added the following sentence in the discussion section: “Other potential cis-interactions including TM3, in addition to the cis-interaction discussed above, have been suggested^{31,34-36}, and these would also be influenced by TM3 bending.” (Ref.31: Gong et al., Mol Biol Cell., 26:4333-46, 2015, Ref.34: Irudayanathan et al., J Phys Chem B., 120:77-88, 2016, Ref.35: Irudayanathan et al., Ann N Y Acad Sci., 1405:131-146, 2017, Ref.36: Piontek et al., Ann N Y Acad Sci., 1397:143-156, 2017)

Minor comments:

11)

Authors: li 113 “While the structures of mCldn15 and mCldn19, which possess alanine at this position, do not show TM3 bending, the TM3 of hCldn4 with proline is certainly bent”

Reviewer: At the “TM3 thenar”, Cldn3 and Cldn4 but not Cldn19 contain a proline. Similarly, TM3 of Cldn3 and Cldn4 but not that of Cldn19 are mentioned to be bent. However, its mentioned that not Cldn3 and Cldn4 but Cldn19 and Cldn4 are positioned similar to the “palm”. This seems to be contradictory. Please discuss/explain.

Response)

It seems strange that the binding position of the C-CPE with Cldn3 differs from that with Cldn4, although Cldn3 and Cldn4 have a proline residue in the TM3 thenar and a bent TM3 helix. As revealed by the P134A structure, however, there is no direct relation between TM3 bending and the C-CPE binding position. The C-CPE binds to two loops, the V1 and V2 regions, in which residues related to C-CPE binding are at the same position as those of mCldn19 and hCldn4 (Extended Data Fig. 4a, b). The V1 and V2 regions are exposed to solvent and, the structures are very flexible. To clarify this, we added a sentence at line 99 as follows: “This different arrangement of the C-CPE is suggested to derive from the flexibility of the V1 and V2 regions, whose structures form loops and are not fixed.”

We also consider that crystal packing is the reason for the binding of Cldn4 to the C-CPE as if the palm wraps around. In the crystal lattice of Cldn4, the C-CPE closely interacts with artificially fused T4L, which has less structural flexibility and was not used in structure analysis of Cldn3, and the C-CPE of adjacent molecules. As a result, the C-CPE is presumably arranged in a direction approaching the β -sheet of Cldn.

Reviewer #2 (Remarks to the Author):

Claudin-family proteins provide the structural basis for the control of paracellular permeability, which is specifically dependent on differential claudin expression. To date only a handful of claudins have been structurally characterized at high resolution. In this manuscript, the authors solved a novel structure for claudin-3 at 3.4 Angstrom resolution. The major finding of the study was that the third transmembrane domain (TM3) of claudin-3 has a key proline residue (P134) that results in a bend that is not present in other claudins harboring a glycine or alanine residue (e.g. claudin-15 and -19). P134G and P134A mutants of claudin-3 caused the TM3 domain to be straight and comparable to claudin-15. The structure determinations were well controlled for potential confounding factors, such as the requirement for CPE to get crystals to form as well as antiparallel interactions between transmembrane domains in the crystals.

Freeze fracture studies of transfected claudin-null cells demonstrated that wt claudin-3 produced tight junction strands at a lower density than those of claudin-3 P134G or P134A, which is a significant observation. A model is proposed that bent vs straight TM3 containing claudins have different cis-packing that leads to altered strand density. However, the model would benefit from additional testing and verification.

Specific comments

Comment 1)

As the authors point out there are several other claudins that have a comparable proline residue - most notably claudin-4 for which the structural of the CPR-claudin-4 complex has also been solved at 3.5 Angstrom resolution. However, in the comparison in Figure 1c it looks as though the bend is different for claudin-4 vs claudin-3. Is this the case? If so, then the authors need to explain why this is the case

Response)

As pointed out by the reviewer, the TM3 of Cldn4 is less bent than the TM3 of Cldn3. We presumed that the crystal packing rendered the TM3 of Cldn4 straighter than that in its natural condition. In the crystal lattice, four Cldn4/C-CPE complexes are tightly bundled in an asymmetric unit, and TM3 bending is hindered by the adjacent complexes in the direction of the bend, as shown in the figure below. As observed in the figure b, Cldn4/C-CPE complexes closely interact with the adjacent artificially fused T4L, and the T4L pushes back to reduce the bending angle of TM3. On the other hand, in the Cldn3 crystal, the asymmetric unit contains two Cldn3/C-CPE complexes in which the TM3 helices of the two Cldn3 molecules, chain A and C, have different interactions (Extended Data Fig. 2g). The different bending angle between chain A and C is considered to be due to the presence or absence of molecules in the bending direction, as described in the figure legend of Extended Data Fig. 8. Therefore, the bending angle observed in crystal lattice tends to be decreased by molecules existing in the direction of the bend, suggesting that the bending angle of chain C of Cldn3 without hindrance by adjacent molecules is more like the native state.

Fig. Crystal packing of hCldn4.

(a) Asymmetric unit of hCldn4/C-CPE. Four complexes are bundled via the TMD of hCldn4. (b) T4L of an adjacent molecule exists in the TM3 bending direction. hCldn4 and C-CPE are colored green and magenta, respectively. T4L of the adjacent molecule is colored gray.

Comment 2)

Also, it is difficult to visualize Figure 1c. Presenting claudin-3, -4, -15 and -19 as individual molecules to highlight bent vs straight TM3 in addition to the the overlay would be extremely helpful (similar to what was done in Extended Figure 4)

Response)

Thank you for your advice, which helped us to clarify the figure. We re-created the figure showing each subtype separately in addition to the overlay. This figure makes it possible to clearly compare the overall structures of the individual subtypes. Along with changing this figure, we reorganized Fig. 1 into two figures as Fig.1 and Fig. 2.

Fig. 2. Comparison with structure-determined subtypes.

(a) Structure-determined Cldn subtypes. mCldn3_{cryst}, hCldn4, mCldn19, and mCldn15 are colored blue, green, yellow, and orange, respectively. C-CPEs paired with mCldn3_{cryst}, hCldn4, and mCldn19 are omitted for clear display of the Cldns. Pro134 of mCldn3 is colored cyan. (b) Superposition of mCldn3_{cryst} and the other structure-determined subtypes.

Comment 3)

p1 "TJ strands among different epithelia have different morphologies, and the TJ strand morphology such as number of strands is suggested to correlate with paracellular permeability". In fact, this is not the case, as demonstrated by Colegio, et al., 2003. Am. J. Physiol. Cell Physiol. 284:C1346-54.

Response)

Yes, Colegio and colleagues showed that swapping the extracellular domain between Cldn2 and Cldn4 changed their barrier or permeation properties without changing the TJ morphology. They revealed that the extracellular region determines the TJ properties, but could not exclude the possibility that TJ morphology controls the degree of the TJ properties, such as barrier strength or degree of selectivity. To evaluate the effect of TJ morphology on the TJ properties, it is necessary to change the TJ morphology without any mutation of the extracellular region. As we explained in this paper, our proline mutation of Cldn3 only introduced one residue to the TM3 helix and not to the extracellular regions. Therefore, it is a safer system for evaluating the effect of TJ morphology on TJ properties.

To clarify, we changed the sentence "TJ strands among different epithelia have different morphologies, and the TJ strand morphology such as number of strands is suggested to correlate with paracellular permeability" to "TJ strands among different epithelia have different morphologies. The relationship between the morphology and properties of TJ strands is well discussed, but still elusive⁸⁻¹⁰. To evaluate the effect of TJ morphology on TJ properties, it is necessary to change the TJ morphology with minimal mutation in the same expression system." (Ref.8: P. Claude, D. A. Goodenough, *J. Cell Biol.* 1973, 58, 390-400, Ref.9: P. Claude, *J. Membr. Biol.* 1978, 39, 219-232, Ref.10: Colegio, et al., 2003, *Am. J. Physiol. Cell Physiol.* 284:C1346-54)

Comment 4)

Also, in the Colegio study, claudin-2 and claudin-4 were found to have dramatically different strand density even though both have the TM3 proline residue comparable to claudin-3. A more thorough literature survey of claudin-specific tight junction strand architecture would strengthen the manuscript as would explaining how TM3 bends in different claudins can lead to different strand architecture.

Response)

Thanks for this comment. We would like to explain more precisely about the determinants of cis-interaction, because we agree that not only TM3 bending, but also the size of the side-chains of the residues forming the cis-interaction pocket or CIN is important for determining strand flexibility, as mentioned on line 239. To clearly state this point, we changed the sentence on line 239 as follows: “The mobility may be determined by the relationship between the sizes of the pocket and CIN, and the TM3 structure is a major determinant of the pocket size, in addition to the kinds of residues comprising the cis-interaction pocket and CIN depending on the Cldn subtype.”

Furthermore, because comprehensive factors, such as the effects of endogenously expressing claudins, are involved in TJ strand formation, we cannot yet exclude the influence of such an uncontrollable factor on TJ strand formation unless the other conditions are the same. In Colegio’s study, they used MDCKII cells expressing considerable endogenous Cldns, so we are afraid that detailed comparison of TJ strand morphology would be difficult in their expression system. Therefore, we excluded the influence of endogenous Cldns by using the BacMam system and HEK293 cells, which do not express endogenous Cldns. We consider that it is difficult to evaluate the strand morphology in other studies that used using different expression systems, so we did not do it in this paper.

Comment 5)

The adhesive assay using transfected SF7 cells is somewhat informative, but is highly sensitive to parameters such as the pipetting frequency (e.g. Figure 4a vs Extended Figure 9d). If trans-interactions are different for wild type vs P134G and P134A claudin-3, this should be detectable by a difference in barrier function (e.g. TER or flux of small molecules) that could be done with transfected mammalian cells. Wild type claudin-3 would also be predicted to have a higher rate of turnover than P134G or P134A claudin-3, which is also testable.

Response)

We are also interested in how the barrier function changes due to the presence or absence of TM3 bending. Structural comparison between mCldn3_{cryst} and mCldn3_{cryst} P134A revealed that the position of the region forming the paracellular pore is affected by TM3 bending (Fig. 3), and therefore it is possible that the barrier function differs between wild-type and mutants. Unfortunately, we must introduce a new measurement technique for evaluating TER or the flux of small molecules, which is difficult to accomplish in a reasonable period. Therefore, we would like to perform an experiment to evaluate the barrier function for our next paper. We agree that the dissociation assay is a sensitive method, but it is a conventional and also nicely reproducible method that has been used in various papers to evaluate adhesion strength (Ozawa et al., PLoS One. 2014, 9, 8; Savignac et al., J Invest Dermatol. 2014, 134, 7,1961-1970). We obtained a statistically significant difference between wild-type and mutants in enough experiments to support the findings from our model that TM3 bending contributes to the adhesivity of TJs. We evaluated the turnover rate with a FRAP assay as described in response to your next comment. Please see the next response for details.

Comment 6)

The model presented in Figure 5 and described in lines 230-246 was not directly tested. If the change in cis interactions occurs when proline is mutated to glycine or alanine, then differences in mobility between wild type and point mutant claudin-3 could be directly detected by a technique such as FRAP. This is feasible considering that pre-existing EGFP-tagged claudin-3 constructs have already been developed and would strengthen the study.

Response)

According to your suggestion, we conducted a FRAP assay. We observed no significant difference in the fluorescence recovery speed or mobile fraction between the wild-type and mutants (see figure below). We interpret this result as indicating only that TM3 bending has no effect on turnover, and believe that it is not correlated with adhesivity or cis-interaction mobility. We conducted this experiment, but we cannot discuss the correlation with adhesion at this time. Therefore, we will not mention this result in the present paper.

Reviewer #3 (Remarks to the Author):

Comment 1)

The structures presented in this work appear to be properly determined and refined. I have a few suggestions that should be addressed.

Response)

We very much appreciate the reviewer's comments regarding the quality of the structure. According to the reviewer's suggestions, we modified the following points. We hope that the improved manuscript clarifies the structural reliability.

First of all, according to your suggestion, we re-evaluated the resolution of the diffraction data. Because the crystals showed strong anisotropy, we submitted our data to the anisotropy server (<https://services.mbi.ucla.edu/anisoscale/>) to estimate the resolution. As mentioned by the reviewer, however, we are afraid that bad Rmerge values make the structures less reliable. Based on CC1/2, the resolution of wild-type, P134G, and P134A was estimated to be 3.6, 4.3, and 3.9 Å, respectively. The main finding of our paper is that the proline residue of TM3 causes a conformational change of the extracellular domain, which is sufficiently reliable under these resolutions. Therefore, to ensure structural reliability, we would like to apply these resolutions to each structure.

Comment 2)

Given the low resolution of the data can the authors comment on what restraints were used in the refinement in the methods section.

Response)

Because of the resolution limit of mCldn3_{cryst} P134G, we only applied rigid body refinement to the model of mCldn3_{cryst} P134G. In the rigid body refinement process, mCldn3_{cryst} P134G was divided into the following three segments: residues of 1-27, 78-133, and 160-200 amino acids (aa) as the transmembrane domain, 28-77 aa as the extracellular domain, and 134-149 aa as the thumb of the TM3 helix. The model was manually rebuilt under the secondary structure restraints referring to the mCldn3_{cryst} P134A model.

We now describe the above restraints in the section of Structure determination.

Comment 3)

As noted in the header of the PDB validation reports that were provided, the report does not correspond to a PDB submitted structure. There is no PDB code listed in the manuscript for the three structures determined in the paper. The authors should complete the PDB submission and provide the official validation report for the PDB processed structures along with a PDB code for each structure. The paper should not be published until this is completed.

Response)

We immediately deposited three structures of mCldn3. The structures of mCldn3 wild-type, P134A, and P134G were assigned as 6AKE, 6AKF, and 6AKG, respectively. These structures were evaluated and certified as reasonable

structures. They are, therefore, ready for publication (HPUB: processing complete, entry on hold until publication).

Comment 4)

comments on Extended Table 1:

The authors should report the appropriate number of significant figures for the Rmerge, Rpim, Mean I(s)I, Completeness and average B factors. For example, a RMerge value of 0.05188 is too many digits for the level of precision. See commentary on appropriate reporting of values (Dauter & Baker, Acta Cryst D, 2007 Mar;63(Pt 3):275). The authors should go through the table and correct these.

Response)

Thank you for the kind suggestions. Following your suggestions, we modified Extended Data Table 1

Comment 5)

The Rmerge value for the highest resolution data shell is quite high in all three structures. The authors should provide the CC1/2 and CC* values for the highest resolution shell to assess whether these reflections are useful in refinements and what the resolution cutoff of the data should be.

Response)

As described above, we re-evaluated the resolution of the data and added a row for CC1/2 in Extended Data Table 1.

Comment 6)

No. of atoms and average B-factors should be reported separately for each chain in the structure.

Response)

We added the number of atoms and average B-factors to Extended Data Table 1, as well as individual values of each chain to Extended Data Fig. 2 (panel h) to show the correspondence with an asymmetric unit. Your meaningful suggestion regarding the average B-factor of each chain helped us demonstrate the flexibility of the TM3 in the P134G mutant. As shown in Extended Data Fig. 2, an asymmetric unit contains two complexes (A/B and C/D) of mCldn3/C-CPE in the case of mCldn3_{cryst} and the P134G mutant, and the two complexes asymmetrically interact with each other in an upside-down manner. The extracellular part of TM3 in chain A approaches the cytosolic part of TM1 in chain C, whereas the extracellular part of TM3 in chain C is exposed to solvent. The B-factor value of each complex is similar in the mCldn3_{cryst} structure. On the other hand, in the P134G mutant, the B-factor of chain C is relatively higher than that of chain A. The higher B-factor of chain C in the P134G mutant would reflect wobbling of the TM3 exposed to solvent. Therefore, this finding supports our idea that the glycine on TM3 destabilizes the helix, leading to wobbling. Additionally, the asymmetric unit of the P134A mutant contains two upside-down complexes (chain A/B/C/D and E/F/G/H) in the same manner as chain A/B/C/D of mCldn3_{cryst}. The values of the B-factors of two Cldn molecules (chain A and C, or E and G) among the upside-down complexes are similar to each other despite the higher B-factor of E/G than A/C. From these comparisons, the TM3 of mCldn3_{cryst} and the P134A mutant have fixed bent and straight forms, respectively, whereas the TM3 of the P134G mutant appears to be straight but is flexible.

Thanks again for your suggestion, which helped us to further clarify our findings.

Appreciating the reviewer's comments, we added the following description at line 137: "The B-factor analysis for each chain in an asymmetric unit also supports this notion, as described below. Comparison of the B-factors between the two Cldn molecules in the upside-down dimer (Extended Data Fig. 2g) revealed that the B-factors of two molecules are similar to each other in the case of mCldn3_{cryst} and mCldn3_{cryst} P134A, whereas the B-factor of chain C is relatively higher than that of chain A only in the case of mCldn3_{cryst} P134G. In a crystal lattice, the extracellular part of TM3 of chain A approaches the cytosolic part of TM1 of chain C, whereas the extracellular part of TM3 of chain C is exposed to solvent. The higher B-factor of chain C in the P134G mutant implies the flexibility of the TM3 exposed to solvent. Therefore, it is considered that the glycine on TM3 destabilizes the helix, leading to wobbling, and that the TM3 of mCldn3_{cryst} and mCldn3_{cryst} P134A have fixed bent and straight forms, respectively."

Comment 7)

Given the low resolution of the data, the authors should provide a Ramachandran plot with distribution of residues and a Molprobit score to assess the quality of the geometry of the structures.

Response)

We apologize that we did not provide a Ramachandran plot in Extended Data Table 1. The new Extended Data Table 1 contains the Ramachandran plot, which we hope will satisfy the reviewer's concern about the reliability of the low-resolution structure.

Additionally, the Molprobit scores of the wild-type, P134G, and P134A structures are 1.43, 1.61, and 1.42, respectively. These scores indicate that these structures are well refined.

Reviewers' comments:

Reviewer #1 (Remarks to the Author):

Nakamura et al. - Morphologic determinant of tight junctions revealed by claudin-3 structures - R1

Comments to the authors:

1. "In the mCldn3 structure, the side-chain of the leucine interacts with the hydrophobic pocket of the C-CPE formed by the tyrosine cluster (see figure below). Therefore, the properties of the side-chain in this position would determine C-CPE affinity".

line 108 "... the interaction in this position determines C-CPE affinity"

This was previously demonstrated and

line 107 "A side-chain of this residue protrudes into a hydrophobic pocket comprising tyrosines of the C-CPE (Extended Data Fig.6b)"

was previously predicted by mutagenesis studies including also claudins in a native membrane environment (Veshnyakova et al., 2012; J Biol Chem. 287(3):1698-708). Discussion of the mCldn3 structure would benefit by referring to this study which was performed without knowledge of a claudin crystal structure.

2. line 186 "Therefore, these findings indicate that the disruption of the cis-interactions by C-CPE ..."

Since direct disruption of cis-interactions by C-CPE was not shown in this study and no supporting reference is provided by the authors "presumed disruption of the cis-interactions by C-CPE" or something similar seems to be more appropriate.

3. Model for TJ strand flexibility

Comment to the authors

Reviewer: "The proposed model is one conceivable explanation of the striking flexibility of most TJ strands. However, the model remains speculative, e.g. since the morphological data are not conclusive. Please consider the recent study of Zhao et al., Communications Biology, 2018 1, 50 which focused on structural determinants of TJ strand flexibility."

Authors: "...Therefore, we do not think that it is appropriate to refer to their arguments in our paper"

It's questionable if the response of the authors to this comment is appropriate. I still think consideration of the Zhao et al study is justified.

Reviewer #2 (Remarks to the Author):

This was a revised manuscript studying the structure of claudin-3, focusing on the ramifications of a bend in TM3 due to a critical proline (P134) on strand architecture. P134G and P134A mutants of claudin-3 caused the TM3 domain to be straight which resulted in claudin-3 forming curved tight junction strands at higher density as opposed to straight strands when expressed in a claudin-null cell line as assessed by freeze fracture EM.

By and large the authors were responsive to the critiques.

In the description of the differential cis-packing model shown in Figure 6 (lines 249-268) the authors refer to a difference in "mobility of the cis-interaction" between wild type and mutant claudin-3. This terminology is confusing, since in this case, "mobility" refers to the proposed capacity for straight TM3 versions of claudin-3 (e.g. P134G) to accommodate multiple cis binding orientations as compared with wild type claudin-3 which is proposed to accommodate fewer cis interaction orientations. It would be helpful to rewrite this section of the manuscript as well as the legend to Figure 6 to clarify this key point.

Also, the authors should consider including the FRAP data shown in the rebuttal as part of the Extended Data set, in that it shows that the rates and extent of recovery are nearly identical for wildtype vs P134G and P134A. These data suggest that despite differences in strand architecture, bulk mobility of claudin-3 (and likely cis binding affinity) is independent of the bend in TM3.

Michael Koval

Reviewers' comments:

Reviewer #1 (Remarks to the Author):

Nakamura et al. - Morphologic determinant of tight junctions revealed by claudin-3 structures - R1

Comments to the authors:

1. “In the mCldn3 structure, the side-chain of the leucine interacts with the hydrophobic pocket of the C-CPE formed by the tyrosine cluster (see figure below). Therefore, the properties of the side-chain in this position would determine C-CPE affinity”.

line 108 “... the interaction in this position determines C-CPE affinity”

This was previously demonstrated and line 107 “A side-chain of this residue protrudes into a hydrophobic pocket comprising tyrosines of the C-CPE (Extended Data Fig.6b)” was previously predicted by mutagenesis studies including also claudins in a native membrane environment (Veshnyakova et al., 2012; J Biol Chem. 287(3):1698-708). Discussion of the mCldn3 structure would benefit by referring to this study which was performed without knowledge of a claudin crystal structure.

Response:

Thank you for your advice relating to references. We would like to refer to the Veshnyakova’s study. Their predictions are consistent with our results that the interactions between the leucine of claudin and the tyrosines of C-CPE are critical for the C-CPE affinity. We refer to this at line 107 and change the sentence as follows: “A side-chain of this residue protrudes into a hydrophobic pocket comprising tyrosines of the C-CPE as previously predicted by a mutagenesis study³⁰...” (Ref. 30: Veshnyakova et al., J Biol Chem. 2012, 287(3):1698-708)

2. line 186 “Therefore, these findings indicate that the disruption of the cis-interactions by C-CPE ...”

Since direct disruption of cis-interactions by C-CPE was not shown in this study and no supporting reference is provided by the authors “presumed disruption of the cis-interactions by C-CPE” or something similar seems to be more appropriate.

Response:

As the reviewer mentioned, the disruption of cis-interactions by C-CPE was not directly indicated but indirectly considered from the comparison of the structures before and after C-CPE binding among different claudin subtypes. To describe more appropriate, we change “indicate” to “presume” in this sentence.

3. Model for TJ strand flexibility

Comment to the authors

Reviewer: “The proposed model is one conceivable explanation of the striking flexibility of most TJ strands. However, the model remains speculative, e.g. since the morphological data are not conclusive. Please consider the recent study of Zhao et al., *Communications Biology*, 2018 1, 50 which focused on structural determinants of TJ stand flexibility.”

Authors: “...Therefore, we do not think that it is appropriate to refer to their arguments in our paper”

It`s questionable if the response of the authors to this comment is appropriate. I still think consideration of the Zhao et al study is justified.

Response:

We reconsidered discussion of the Zhao’s study in our paper. Although we were skeptic about their cis-interfaces because they evaluated the stability of only a dimer and did not discuss the possibility of polymerization, their ideas that laterally-rotated cis-interactions contribute the strand flexibility are close to our ideas that the permissible orientation range depending on the cis-interaction pocket size determines the strand flexibility. We add the discussion about it at line 274 as follows: “Our study proposes that TM3 bending alters the cis-interaction pocket size and the permissible orientation range of cis-interaction is one important factor determining TJ strand flexibility. The results recently reported by molecular dynamics simulation predicted that laterally-rotated cis-interactions are necessary for the TJ strand flexibility³⁵, which were similar concepts to our model for TJ strand flexibility.” (Ref 35. Zhao et al., *Communications Biology*, 2018, 1, 50)

Reviewer #2 (Remarks to the Author):

This was a revised manuscript studying the structure of claudin-3, focusing on the ramifications of a bend in TM3 due to a critical proline (P134) on strand architecture. P134G and P134A mutants of claudin-3 caused the TM3 domain to be straight which resulted in claudin-3 forming curved tight junction strands at higher density as opposed to straight strands when expressed in a claudin-null cell line as assessed by freeze fracture EM.

By and large the authors were responsive to the critiques.

1) In the description of the differential cis-packing model shown in Figure 6 (lines 249-268) the authors refer to a difference in "mobility of the cis-interaction" between wild type and mutant claudin-3. This terminology is confusing, since in this case, "mobility" refers to the proposed capacity for straight TM3 versions of claudin-3 (e.g. P134G) to accommodate multiple cis binding orientations as compared with wild type claudin-3 which is proposed to accommodate fewer cis interaction orientations. It would be helpful to rewrite this section of the manuscript as well as the legend to Figure 6 to clarify this key point.

Response:

Thank you for suggestion relating to the description in the model for TJ strand flexibility. As your suggestion, "mobility" is confusing and does not exactly describe our idea that the cis-interaction pocket size determines permissible orientation range of cis-interaction. Therefore, we rewrite this section (lines 256-265) as follows: "The narrow pocket would tightly grasp the hydrophobic residue of the CIN of the adjacent Cldn by sandwiching from the vertical direction, and restrict the orientation of the cis-interaction, whereas the wide pocket would loosely catch the CIN enabling wide orientation range. The permissible orientation range of cis-interaction may be determined by the relationship between the sizes of the pocket and CIN, and the TM3 structure is a major determinant of the pocket size, in addition to the kinds of residues comprising the cis-interaction pocket and CIN depending on the Cldn subtype (Extended Data Fig. 5). The permissible orientation range of cis-interaction would influence TJ strand flexibility. As the result, TJ strands with little flexibility consisting of orientation-restricted cis-interactions (e.g. wild-type mCldn3) represent the linear pattern, whereas highly flexible TJ strands consisting of cis-interactions with widely permissible orientation (e.g. mCldn3 P134G or P134A) represent the curvy pattern." We also change the legend of Figure 6 in the same way.

2) Also, the authors should consider including the FRAP data shown in the rebuttal as part of the Extended Data set, in that it shows that the rates and extent of recovery are nearly identical for wildtype vs P134G and P134A. These data suggest that despite differences in strand architecture, bulk mobility of claudin-3 (and likely cis binding affinity) is independent of the bend in TM3.

Response:

As the reviewer's comment, we recognize that the FRAP data showed an important fact indicating that the fluorescence recovery rates among wild type vs mutants are identical and so molecular mobility of claudin-3 is independent of the TM3 bending. We add the results of FRAP assay as Extended Data Fig. 10. Along with this, we add

the experimental method of FRAP assay in the method section, and describe the results at line 265 as follows: “The FRAP assay showed that the rates and extent of fluorescence recovery are identical among wild-type mCldn3 and P134G or P134A mutant (Extended Data Fig. 10). These data suggest that despite differences in strand architecture, bulk mobility of mCldn3 (and likely cis binding affinity) is independent of the TM3 bending.”

Extended Data Fig. 10. FRAP dynamics of mCldn3 on mammalian cells

(a) FRAP analysis of EGFP-fused mCldn3 (mCldn3 WT, P134G or P134A) stably expressed in SF7 cells. Representative high-magnification images of TJ segments at the indicated time points after photobleaching are shown. Bar, 5 μ m. (b) Quantification of FRAP analysis of EGFP-fused mCldn3 (mCldn3 WT, P134G or P134A) stably expressed in SF7 cells. The half-time ($t_{1/2}$) of fluorescence recovery for mCldn3 WT, P134G or P134A are 6.4±0.6, 7.7±0.8, or 7.5±0.6 sec, respectively. The mobile fractions for mCldn3 WT, P134G or P134A are 62±1, 61±1, or 65±2 %, respectively. The $t_{1/2}$ of fluorescence recovery and mobile fractions show no significant differences among mCldn3 WT, P134G and P134A. Results from three distinct clones are shown as means \pm SEM ($n \geq 15$ /group). P-values were calculated using a two-tailed independent t-test, and $P < 0.05$ was considered significant.

Methods:

FRAP analysis

FRAP analysis was performed according to the previously described method⁴⁹. Briefly, the confluent cells on glass bottom dishes with a 3.5-cm diameter (AGC TECHNO GLASS CO., LTD.) were cultured for 3 days. The cells were pre-incubated in FluoroBrite DMEM Media (Thermo Fisher Scientific K.K.) with 10% fetal bovine serum, 2mM L-glutamine, and 10 μ M forskolin for 6 hours, and processed for the FRAP analysis. FRAP analysis was performed by the LSM880 confocal laser scanning microscopy with the Plan-Apochromat 63 \times 1.4 Oil DIC M27 objective lens (Carl Zeiss Japan). A heated chamber was used to keep the temperature at 37°C and maintain the CO₂ in the chamber at 5%. The images were collected at 1 frame/sec with the following parameters: resolution, 512 \times 512 pixels; pinhole, 1.43 airy unit; excitation wavelength, 488nm; laser transmission, 1.0 %. A targeted region was bleached using 100% laser transmission at 488 nm (1 pulse). The fluorescence intensity was normalized to the prebleach intensity using ImageJ software (National Institutes of Health, Bethesda, MD).